# DEEP CONFIDENT STEPS TO NEW POCKETS: STRATEGIES FOR DOCKING GENERALIZATION

**Gabriele Corso**[*1], **Arthur Deng**[*2], **Nicholas Polizzi**[3], **Regina Barzilay**[1], **Tommi Jaakkola**[1]

[1]CSAIL, Massachusetts Institute of Technology, [2]University of California, Berkeley,
[3]Dana-Farber Cancer Institute and Harvard Medical School

## ABSTRACT

Accurate blind docking has the potential to lead to new biological breakthroughs, but for this promise to be realized, docking methods must generalize well across the proteome. Existing benchmarks, however, fail to rigorously assess generalizability. Therefore, we develop DOCKGEN, a new benchmark based on the ligand-binding domains of proteins, and we show that existing machine learning-based docking models have very weak generalization abilities. We carefully analyze the scaling laws of ML-based docking and show that, by scaling data and model size, as well as integrating synthetic data strategies, we are able to significantly increase the generalization capacity and set new state-of-the-art performance across benchmarks. Further, we propose CONFIDENCE BOOTSTRAPPING, a new training paradigm that solely relies on the interaction between diffusion and confidence models and exploits the multi-resolution generation process of diffusion models. We demonstrate that CONFIDENCE BOOTSTRAPPING significantly improves the ability of ML-based docking methods to dock to unseen protein classes, edging closer to accurate and generalizable blind docking methods.

## 1 INTRODUCTION

Understanding how small molecules and proteins interact, a task known as molecular docking, is at the heart of drug discovery. The conventional use of docking in the industry has led the field to focus on finding binding conformations when restricting the search to predefined pockets and evaluating these on a relatively limited set of protein families of commercial interest. However, solving the general blind docking task (i.e. without pocket knowledge) would have profound biological implications. For example, it would help us understand the mechanism of action of new drugs to accelerate their development [Schottlender et al., 2022], predict adverse side-effects of drugs before clinical trials [Luo et al., 2018], and discover the function of the vast number of enzymes and membrane proteins whose biology we do not yet know [Yi et al., 2015]. All these tasks critically require the docking methods to generalize beyond the relatively small class of well-studied proteins for which we have many available structures.

Existing docking benchmarks are largely built on collections of similar binding modes and fail to rigorously assess the ability of docking methods to generalize across the proteome. Gathering diverse data for protein-ligand interactions is challenging because binding pockets tend to be evolutionarily well-conserved due to their critical biological functions. Therefore, a large proportion of known interactions fall into a relatively small set of common binding modes. Moreover, human biases in the collection of binding conformational data further compromise the representativeness of existing benchmarks. To address these problems, we propose DOCKGEN, a new benchmark that aims to test a method's ability to generalize across protein domains. With DOCKGEN, we show that existing machine learning-based docking methods poorly predict binding poses on unseen binding pockets.

With this new benchmark, we analyze the scaling laws of DIFFDOCK, the state-of-the-art ML-based blind docking method, with respect to the size of both its architecture and its training data. The results show that increasing both data and model can give significant generalization improvements. Further, we devised and integrated a synthetic data generation strategy based on extracting side chains as ligands from real protein structures. Putting these together, our new DIFFDOCK-L

---

*Equal contribution. Correspondence to gcorso@mit.edu

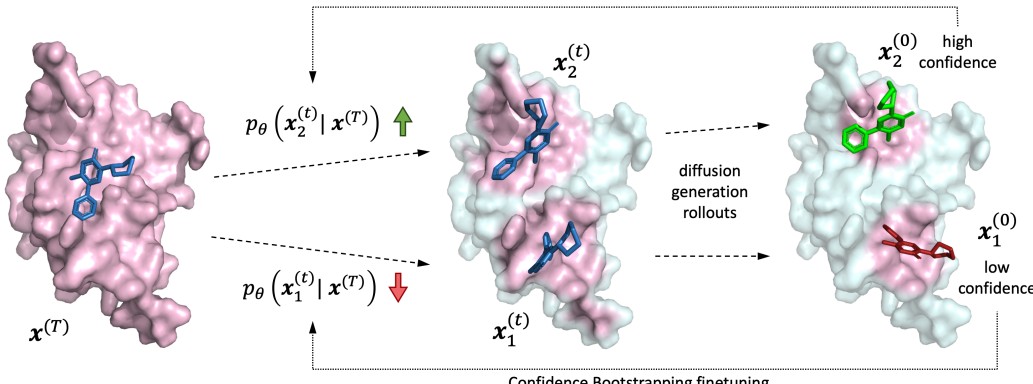

Figure 1: Visual representation of the CONFIDENCE BOOTSTRAPPING training scheme. The dashed lines represent the reverse diffusion generation rollouts that the model executes. The dotted lines illustrate the bootstrapping feedback from the confidence model that is used to update the likelihood of the early diffusion steps by changing the weights of the score model. The pink regions of the protein represent areas to where the docking algorithm is still attending, which starts from being the whole protein and then gradually narrows to the local environment around the current pose.

increases the top ML-based DOCKGEN performance from 7.1% to 22.6% setting a new state-of-the-art. However, with the current data and computing resources available today, this trend alone might not be sufficient to fully bridge this generalization gap.

To move beyond this challenge, we propose CONFIDENCE BOOTSTRAPPING, a novel self-training scheme inspired by Monte Carlo tree-search methods, where we fine-tune directly on protein-ligand complexes from unseen domains without access to their structural data. The fine-tuning is enabled by the interaction between a diffusion model rolling out the sampling process and a confidence model assigning confidence scores to the final sampled poses. These confidence scores are then fed back into the early steps of the generation process (see Figure 1 for a visual representation). This process is iterated to improve the diffusion model's performance on unseen targets, effectively closing the generalization gap between the diffusion model and the confidence model.

We test CONFIDENCE BOOTSTRAPPING on the new DOCKGEN benchmark by fine-tuning a small and efficient version of DIFFDOCK on individual protein clusters. In each of these clusters, within the first few iterations of bootstrapping, the diffusion model is pushed to generate docked poses with increasingly high confidence. This increased confidence also translates into significantly higher accuracy with the fine-tuned models improving from 9.8% to 24.0% success rate overall, and above 30% in half of the protein domains.

## 2 RELATED WORK

**Search-based docking**   Due to its importance in biological research and drug discovery, molecular docking has for decades been a central challenge for the computational science community [Halgren et al., 2004; Jain, 2003; Thomsen & Christensen, 2006]. Originally, most techniques followed the search-based paradigm, which is still prevalent today. These methods consist of a scoring function and an optimization algorithm. The latter searches over thousands or millions of different conformations, which are passed to the scoring function that determines their likelihood/goodness. While these methods tend to show relatively robust generalization across protein classes, they are significantly affected by the size of the search space, which grows exponentially as the ligand gets larger or as assumptions are removed (e.g. receptor rigidity).

**ML-based docking**   Researchers have recently tried to move beyond the search-based paradigm and directly generate poses with deep learning models. The first attempts [Stärk et al., 2022; Lu et al., 2022] framed the docking task as a regression problem; this showed significant improvements in runtime but did not reach the accuracy of search-based methods. Corso et al. [2022] proposed DIFFDOCK, a generative model based on the diffusion modeling framework that is trained to sample

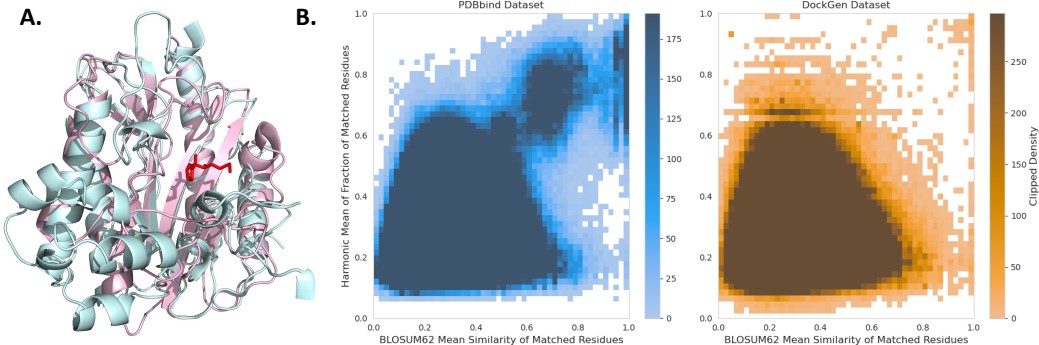

Figure 2: **A.** An example of the superimposition of the pockets of two proteins in PDBBind, 1QXZ in pink and 5M4Q in cyan, that share a very similar binding pocket structure (a bound ligand is shown in red), but have only 22% sequence similarity. While sequence similarity splits would classify them in separate clusters, our approach correctly identifies that the binding domain of these two proteins is the same. **B.** Comparison of binding sites in train vs test set for both PDBbind and DOCKGEN datasets. BLOSUM62 and harmonic mean similarity metrics (more details in Appendix A.2) have a maximum of 1 (most similar) and a minimum of 0 (least similar). The densities are clipped at 1% of the maximum value for both datasets to emphasize contamination. Every binding site in the train set was compared to every binding site in the test set showing significantly higher train-test similarity in the PDBBind dataset compared to the DOCKGEN dataset.

docked ligand poses. In particular, DIFFDOCK uses a diffusion model to sample a small number of possible poses that are then passed to a confidence model that ranks and assigns a score to each.

**Blind docking benchmarks** The majority of previous ML-based methods used the PDBBind dataset [Liu et al., 2017], a curated set of protein-ligand crystallographic structures from PDB [Berman et al., 2003], to train and test models. In particular, they adopted a time-based split of the dataset where structures that were resolved before 2019 went into training and validation, and those from 2019 formed the test set. Stärk et al. [2022] and others also evaluate the performance on a reduced set of proteins with different UniProt-ID [Consortium, 2015] compared to those in the training set. Here, while ML methods show a drop in performance, they remain in line with search-based techniques [Corso et al., 2022]. Similarly, concurrent works [Masters et al., 2023; Buttenschoen et al., 2024] define new splits or benchmarks based on global sequence similarity.

## 3 THE DOCKGEN BENCHMARK

We argue that the existing approaches used to evaluate the ML docking methods fall short in analyzing their generalization capacity to different parts of the proteome. Binding pockets, due to their importance to many biological mechanisms, are often among the most well-conserved regions in proteins. Therefore, just looking at the UniProt-ID of a protein or its global sequence similarity often leads to proteins in the train and test sets that have the same underlying pocket. Figure 2-A shows an example of such failures, where two proteins, even with only 22% sequence similarity (30% is often used as cutoff), share very similar binding pockets.

To better detect these cases we delve one level deeper into the organization of proteins and look at protein domains. Protein domains are the structural and functional units that compose proteins. Very similar domains can appear in different sequences but have similar local structural characteristics. Therefore, by looking at the protein domains where ligands bind, we can form a more granular classification of the protein-ligand interactions.

To classify each domain, we used the ECOD [Cheng et al., 2014] classification. This clustering divides the 17k complexes from PDBBind before 2019, which have been used for training and validation of previous ML models, into 487 clusters. The remaining data from 2019, from which the test set was generated, presents only 8 additional clusters composed of a total of 15 complexes. This clustering approach is very different from that taken by other methods based on global sequence similarity.

Table 1: Top-1 RMSD performance of different methods on the PDBBind and DOCKGEN benchmarks. Runtimes were computed as averages over the PDBBind test set. * run on CPU. Med. indicates the median RMSD. Ex. refers to the level of exhaustiveness of the search in case this was increased above the default value. †more details in Section 5.

| Method | PDBBind %<2Å | PDBBind Med. | DOCKGEN-full %<2Å | DOCKGEN-full Med. | DOCKGEN-clusters %<2Å | DOCKGEN-clusters Med. | Average Runtime (s) |
|---|---|---|---|---|---|---|---|
| SMINA | 18.7 | 7.1 | 7.9 | 13.8 | 2.4 | 16.4 | 126* |
| SMINA (EX. 64) | 25.4 | 5.5 | 10.6 | 13.5 | 4.7 | 14.7 | 347* |
| P2RANK+SMINA | 20.4 | 4.3 | 7.9 | 14.1 | 1.2 | 16.4 | 126* |
| GNINA | 22.9 | 7.7 | 14.3 | 15.2 | 9.4 | 14.5 | 127 |
| GNINA (EX. 64) | 32.1 | 4.2 | 17.5 | 8.1 | 11.8 | 6.2 | 348 |
| P2RANK+GNINA | 28.8 | 4.9 | 13.8 | 16.2 | 4.7 | 15.3 | 127 |
| EQUIBIND | 5.5 | 6.2 | 0.0 | 13.3 | 0.0 | 13.3 | **0.04** |
| TANKBIND | 20.4 | 4.0 | 0.5 | 11.6 | 0.0 | 11.1 | 0.7 |
| DIFFDOCK (10) | 35.0 | 3.6 | 7.1 | 6.8 | 6.1 | 6.0 | 10 |
| DIFFDOCK (40) | 38.2 | 3.3 | 6.0 | 7.3 | 3.7 | 6.7 | 40 |
| DIFFDOCK-L† (10) | **43.0** | **2.8** | **22.6** | **4.3** | **27.6** | **3.7** | 25 |
| DIFFDOCK-S + C.B.† (10) | - | - | - | - | 24.0 | 3.8 | 2.8 |

To obtain a more sizable test set without retraining the models on a reduced set, we turn to the Binding MOAD dataset [Hu et al., 2005]. Similar to PDBBind, Binding MOAD is a curated collection of protein-ligand complexes from the PDB. However, due to its different filtering and requirements (e.g. no requirement for known affinity), it contains a set of 41k complexes partially disjoint from PDBBind. These come from 525 ECOD clusters, 346 of which are in common with PDBBind, and 179 of which are not present in PDBBind.

To generate the validation and test datasets of the new benchmark, we randomly divide these remaining clusters in two and then apply a number of further filtering steps (more details in Appendix A). In particular, we exclude protein-ligand complexes with multiple ligands interacting in the same pocket (i.e. no additional bound cofactors). We also remove metals, crystal additives, and large molecules with more than 60 heavy atoms. To maintain a chemical balance we only keep up to 5 complexes with the same ligand in the validation and test datasets. This leaves us with 141 complexes in the validation and 189 complexes in the test set. A careful analysis of the binding-site similarity of the different datasets highlights the vast improvement brought by DOCKGEN in terms of binding site generalization (Figure 2.B and Appendix A.2).

We then run a number of baselines considered to be the state-of-the-art open-source or open-access models: for search-based methods, SMINA [Koes et al., 2013] and GNINA [McNutt et al., 2021], while for ML methods, EQUIBIND [Stärk et al., 2022], TANKBIND [Lu et al., 2022] and DIFFDOCK [Corso et al., 2022]. Since search-based methods have been shown to improve their blind docking performance by first selecting a pocket with a pocket finder method like P2RANK [Krivák & Hoksza, 2018], we also report these performances.

Previous ML methods significantly underperform in this new benchmark (Table 1), and their performances are only a fraction of those that they have in the time-split complexes from the PDBBind test set, with regression methods having nearly no success. On the other hand, search-based methods have a significantly lower drop in performance, but even when increasing the exhaustiveness of their search, they are not able to find the true pose in the vast majority of cases, motivating the need for the development of more accurate methods.

## 4 CONFIDENCE BOOTSTRAPPING

Docking, along with other structural biology problems like protein folding, has been commonly treated as an NP-hard combinatorial optimization problem [Thomsen, 2007; Mukhopadhyay, 2014]. Although the success of ML methods such as AlphaFold2 [Jumper et al., 2021] has demonstrated that deep learning methods can significantly shorten the search in practice. The NP perspective suggests a useful insight into the problem: it is easier to check that a pose is good than to generate a good pose. This perspective points towards the exploration of new self-training-based strategies where feedback from a discriminative model is used to update the generative model exploring the

space of conformations and help the latter generalize to protein domains where we have no data for ground truth poses.

## 4.1 BACKGROUND

**Diffusion models** Let $p(\mathbf{x})$ be some data distribution of interest. Score-based diffusion generative models [Song et al., 2021] are trained to sample from $p$ by defining a continuous diffusion process $d\mathbf{x} = f(\mathbf{x}, t)dt + g(t)d\mathbf{w}$, where $\mathbf{w}$ represents the Wiener process that transforms the data distribution in a simple prior, and learns to reverse such a process. This is enabled by the existence of a corresponding reverse diffusion process which can be expressed by the SDE $d\mathbf{x} = [f(\mathbf{x}, t) - g(t)^2 \nabla_{\mathbf{x}} \log p_t(\mathbf{x})]dt + g(t)d\mathbf{w}$ where $p_t(\mathbf{x})$ is the likelihood of the evolving data distribution. To run this reverse diffusion equation, we need to learn to approximate the score of the evolving data distribution $\mathbf{s}_\theta(\mathbf{x}, t) \approx \nabla_{\mathbf{x}} \log p_t(\mathbf{x})$. This is achieved by optimizing the parameters $\theta$ via the denoising score matching loss:

$$\theta^* = \arg\min_\theta \left[ \mathbb{E}_{t \sim U[0,T]} \left\{ \lambda(t) \mathbb{E}_{\mathbf{x}^{(0)} \sim p_{\text{train}}} \mathbb{E}_{\mathbf{x}^{(t)} | \mathbf{x}^{(0)}} \left[ \| s_\theta(\mathbf{x}^{(t)}, t) - \nabla_{\mathbf{x}^{(t)}} \log p_{0t}(\mathbf{x}^{(t)} | \mathbf{x}^{(0)}) \|_2^2 \right] \right\} \right]$$

where $U$ refers to the uniform distribution, $\lambda(t)$ is a positive weighting function and $p_{0t}$ the transition kernel of the forward diffusion process. One view of diffusion models is via the lens of their iterative denoising generation ladder which, at every step, exponentially reduces the size of candidate poses' posterior distribution (progressively higher resolution), a perspective we will use to motivate our approach. Diffusion models were also generalized to compact Riemannian manifolds [De Bortoli et al., 2022], a formulation that is particularly well suited for scientific applications where the main degrees of freedom can be well represented by actions on a low-dimensional Riemannian manifold [Corso, 2023]. This idea underlies DIFFDOCK and other recent advances in computational sciences [Jing et al., 2022; Watson et al., 2023].

**Self-training methods** Self-training refers to a range of techniques that have been employed in several different ML application domains where labels predicted by some model on unlabelled data are used for training. For example, in the setting of image classification, Xie et al. [2020] used unlabelled images to improve a classifier by first generating labels from the clean image with the current classifier version, and then training the classifier to make the same prediction on noisy versions of the same image. This method was taken as inspiration for the self-distillation technique used by AlphaFold2 [Jumper et al., 2021], where after a first training iteration, predicted structures satisfying a certain residue pairwise distance distribution were used for a second iteration of model training.

In the realm of generative models, McClosky et al. [2006] used the labels predicted by a discriminative reranker to select the best parses generated by a generative parser and add them to the training set. Jin et al. [2021] took a similar approach for antibody optimization via the feedback of a neutralization predictor. Finally, Generative Adversarial Networks (GANs) [Goodfellow et al., 2014] also use the feedback from a discriminator to train a generative model. However, in GANs one relies on having in-distribution data to jointly train the discriminator.

## 4.2 METHOD

These existing self-training methods do not, however, directly exploit the structure of the generative process they optimize. Moreover, they often fail if the initial generator has a low signal-to-noise ratio. Iterating on such generators amplifies the initial errors [McClosky et al., 2006]. We argue that diffusion models, because of their multi-resolution structure, offer a unique opportunity to more precisely target the effect of self-training and avoid error amplification.

A large challenge for the diffusion model is that in the first steps of the reverse diffusion process, the model needs to determine both the pocket and approximate pose of the ligand, without having a clear view of how well that pose will fit in the pocket. If the model finds that the pocket does not fit the ligand adequately after the ligand has been partially docked, the model can not backtrack its decisions[1] or learn from its mistakes.

---

[1] Note that in theory at every step the conditional distribution learned by the diffusion model spans the whole search space; however, in practice the model learns, based on the prior, to not to look for a conformation beyond a distance that is much larger than the current noise level.

We introduce CONFIDENCE BOOTSTRAPPING, a training mechanism that refines a diffusion generator based on feedback from a confidence model. The diffusion model is used to "roll out" the reverse diffusion process, generating poses that are then scored with the confidence model. These scores are used to inform how to update the parameters of the early steps of the diffusion process so that the model will generate more poses close to those with high confidence (see a graphical representation in Figure 1). This process is then repeated for several steps.

There are several reasons why we hypothesize that the diffusion model would be able to learn from the feedback of the confidence model. As discussed above, while generating a ligand pose for a new protein is difficult, testing whether a pose is satisfactory is a local and simpler task that, we believe, can more easily generalize to unseen targets. Intuitively, the more difficult the underlying task the more a model is likely to overfit, capturing some spurious correlations in the data instead of the underlying signal. Similarly, early steps of the reverse diffusion process that have to decide the most likely among the different pockets and poses will struggle more than the late steps where the remaining posterior is significantly more localized. Training the early steps of the reverse diffusion based on the confidence outcomes of the rollouts, CONFIDENCE BOOTSTRAPPING is able to exploit the multi-resolution structure of diffusion models and close the generalization gap between the different diffusion steps.

From this perspective, our procedure of (1) rolling out the steps of the generation process, (2) evaluating the success at the end, and (3) feeding back information from the end to the initial steps resembles the Reinforcement Learning (RL) algorithm used to master games such as Go [Silver et al., 2016]. Instead of score and confidence models, they use policy and value networks, and the diffusion process is replaced by a Monte Carlo tree-search. More generally, our problem can be loosely seen as an RL problem, where confidence is the reward. We discuss this connection in more detail in Appendix D.

Although it adds complexity to the training process, CONFIDENCE BOOTSTRAPPING is particularly well suited for the molecular docking task. Firstly, as discussed in Section 3, the limited amount of training data and its lack of diversity make alternative training methods critical for the success of blind docking. Furthermore, CONFIDENCE BOOTSTRAPPING can leverage information from very large affinity datasets such as BindingDB [Liu et al., 2007]. Exploiting this binding data is largely unexplored for the docking task. Finally, docking screens are usually run on a very large number of complexes (up to tens of billions [Gorgulla et al., 2020]) using a restricted set of proteins. Therefore, any time that one would spend fine-tuning the docking algorithm for the restricted set would be largely amortized throughout the screen.

### 4.3 FORMALIZATION

We now formalize the CONFIDENCE BOOTSTRAPPING training routine. For simplicity, we will present it based on the diffusion model formulation in Euclidean space. However, note that this directly applies to the Riemannian diffusion formulation [De Bortoli et al., 2022] used by DIFFDOCK.

Let $p_\theta(\mathbf{x}; d)$ be the probability distribution of poses learned by the diffusion model with score $s_\theta(\mathbf{x}^{(t)}, t; d)$ where $d$ is the known information about the complex (e.g, protein and molecule identity). Let $c_\phi(\mathbf{x}, d)$ be the output of the confidence model, and let $D = \{d_1, d_2, ...\}$ be a set of known binders (e.g. from BindingDB) for the target distribution of interest.

CONFIDENCE BOOTSTRAPPING consists of $K$ iterations where at each iteration $i$ the score model weights $\theta$ are updated based as following optimization:

$$\theta^{i+1} = \arg\min_\theta \left[ \mathbb{E}_{t \sim U[0,T]} \left\{ \lambda(t) \, \mathbb{E}_{\mathbf{x}^{(0)}, d \sim p_{\text{train}}} \mathbb{E}_{\mathbf{x}^{(t)} | \mathbf{x}^{(0)}} \left[ \|s_\theta(\mathbf{x}^{(t)}, t; d) - \nabla_{\mathbf{x}^{(t)}} \log p_{0t}(\mathbf{x}^{(t)} | \mathbf{x}^{(0)}) \|_2^2 \right] \right. \right.$$

$$\left. \left. + \lambda'(t) \, \mathbb{E}_{\mathbf{x}^{(0)}, d \sim p_{\theta^i, \phi}} \mathbb{E}_{\mathbf{x}^{(t)} | \mathbf{x}^{(0)}} \left[ \|s_\theta(\mathbf{x}^{(t)}, t; d) - \nabla_{\mathbf{x}^{(t)}} \log p_{0t}(\mathbf{x}^{(t)} | \mathbf{x}^{(0)}) \|_2^2 \right] \right\} \right]$$

where $\theta^0$ are the weights of the pretrained diffusion model (if not trained from scratch), and $p_{\theta,\phi}(\mathbf{x}, d) \propto p_\theta(\mathbf{x}; d) \, \exp[c_\phi(\mathbf{x}, d)]$.

Each of these iterations $i \in [0, K)$ is achieved by performing a rollout stage, followed by an update stage. During the rollout stage, we first sample $d$ from $D$, then sample points from $p_{\theta^i}(\cdot, d)$, forming a buffer $B = [(x_1, d_1), ...]$. During the update stage, a fixed number of stochastic gradient descent steps are performed where half of the fine-tuning complexes are taken from the training dataset (first

half of the objective) and half are taken from the buffer $B$ (second half). In particular, to approximate samples from $p_{\theta^i, \phi}(\mathbf{x}, d)$, the elements $(\mathbf{x}, d)$ of $B$ are sampled with probabilities proportional to $\exp[c_\phi(\mathbf{x}, d)]$.

Using different $\lambda$ and $\lambda'$ allows us to take advantage of the multiresolution decomposition of diffusion models and direct the bootstrapping feedback principally to update the initial steps of the reverse diffusion. The samples taken from the combination of diffusion and confidence models are likely to be too noisy to provide fine-grained guidance for small $t$. To prevent the model from forgetting how to perform the final steps, we still take samples from the training dataset and give them a larger weight for small $t$. Further details on the implementation and optimization of this routine can be found in Appendix C.

## 5 EXPERIMENTS

### 5.1 ANALYZING DOCKING SCALING LAWS

Empirically, we first analyze the effect that scaling the training data and the model size has on the generalization capacity of DIFFDOCK. This analysis is important to evaluate the potential impact of community efforts to increase the amount of available high-quality training data and develop large-scale ML docking models.

**Increasing the training data** We investigate how much the addition of more data, within the protein domains that the model already sees, can help with generalization outside of those domains. This is an important question because in recent years there has been a call from the research community for pharmaceutical companies to release the large number of crystallography structures they possess. We test this question by including in the training data all MOAD complexes from the same binding protein domains as those seen in PDBBind training and validation sets and released before 2019 (to maintain the validity of the benchmarks). After filtering out duplicates and non-conformant ligands, this increases the number of training data points by approximately 52%.

**Van der Mer-inspired docking augmentation** Additional training data from MOAD contributes modestly to pocket diversity, as the extra data points lie within the same protein domain clusters. To increase pocket diversity, we design a novel auxiliary training task based on the generation of synthetic docked poses using protein sidechains as surrogate ligands. We take inspiration from the concept of a van der Mer (vdM), which has been used successfully in the design of proteins that bind target ligands [Polizzi & DeGrado, 2020]. A van der Mer is an amino acid that interacts with another amino acid that is distant in the 1D protein sequence, which can closely approximate a noncovalent protein-ligand interaction. In a given protein crystal structure, we select a sidechain with a large number of protein contacts distant in sequence. The interacting amino acids are assigned as the "binding pocket" for the chosen sidechain. We remove the coordinates of the "ligand" residue and its sequence-local neighbors from the protein to generate the new target complex (more details on how exactly these are chosen can be found in Appendix B.2).

The advantage of these synthetic protein-ligand complexes is that they are myriad and easy to compute, since any (even unliganded) protein structure can be used to generate such examples. Thus, we can potentially dramatically increase the structural and chemical diversity of binding domains and pockets. This diversity could help the model understand the chemical and geometrical environments of different pockets. The drawbacks are that these synthetic complexes are of unknown affinity (many could be weak binders), and the chemical diversity of ligands is limited to the 20 amino acids.

**Increasing the model size** Further, we evaluate how the generalization ability of the docking models changes when scaling their capacity. The relationship between model capacity and generalization has been a topic of significant study in the machine learning community. On one hand, the traditional learning theory perspective suggests that once the model overfits to some amount of training data increasing the model size will only likely make the overfitting worse. On the other hand, recent evidence for deep learning models has shown the opposite behavior with "overparameterized" models increasing their ability to generalize when given more capacity [Belkin et al., 2019]. We compare score models of the size of the original DIFFDOCK ($\sim$20M parameters) with others of smaller ($\sim$4M) and larger ones ($\sim$30M).

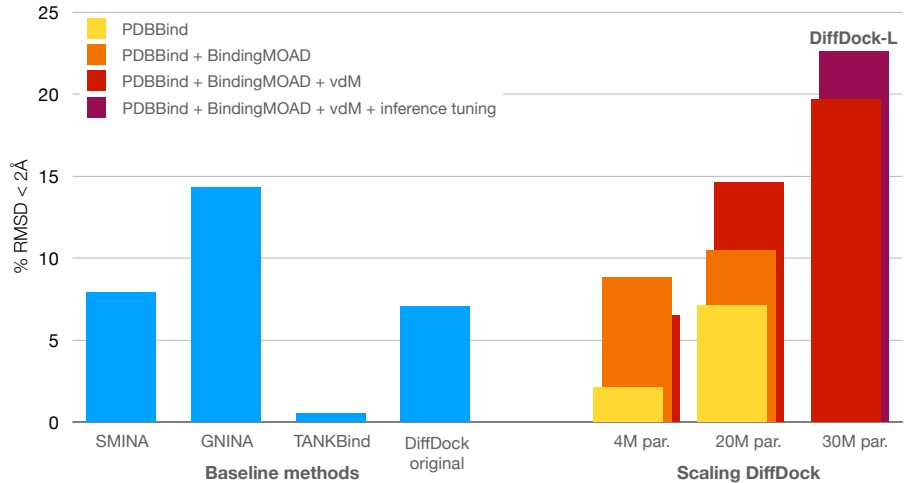

Figure 3: Analysis of the scaling laws of DiffDock when measuring its ability to generalize to unseen protein domains. *par* indicates the number of parameters and the different colors indicate different training sets and augmentations. For the 30M architecture, only one model was trained due to its expensive training cost. Inference tuning refers to tuning various reverse diffusion hyperparameters at inference time.

**Experimental results**   The results of these experiments, shown in Figure 3, highlight a clear trend of improvement both when increasing the training data and when increasing the size of the model. The vdM augmentation strategy also seems to provide some improvements when scaling to larger model sizes. Combining these components we develop DIFFDOCK-L, which we release publicly[2]. DIFFDOCK-L improves the ML-docking performance on DOCKGEN from 7.1% to 22.6%, even outperforming the best search-based method (17.5%). DIFFDOCK-L also achieves state-of-the-art blind docking performance on both PDBBind (see Table 1) and PoseBusters test sets (see Appendix E.1). Overall we believe that the analysis of these trends may prove useful for many practitioners and to support the effort of the community in developing even larger open-source datasets and models.

## 5.2   CONFIDENCE BOOTSTRAPPING

We test[3] CONFIDENCE BOOTSTRAPPING on the new DOCKGEN benchmark, where we fine-tune a model on each protein domain cluster. In particular, we use DIFFDOCK-S, the 4M parameters version of DIFFDOCK introduced in Appendix C.2. For computational feasibility, we use clusters with at least 6 complexes and restrict the test set to 8 separate clusters (5 for validation) for a total of 85 complexes, which compose the DOCKGEN-clusters subset.

As can be seen from Figure 4-A, for most of the clusters, the median confidence of the predicted structures increases along the fine-tuning. Critically, Figure 4-B and 4-C show that also the accuracy of most clusters significantly improves over the course of the bootstrapping process. In Figure 4-D, we plot the average performance across all clusters in comparison to that of the baselines. From this, we see that, in DOCKGEN-clusters, CONFIDENCE BOOTSTRAPPING considerably raises the baseline DIFFDOCK-S's performance going from 9.8% to 24.0% and doubles that of the traditional search-based methods even when run with high exhaustiveness.

The analysis becomes even more interesting when looking into the evolution of the performance in the individual clusters. In half of the clusters, the model is able to reach top-1 RMSD < 2Å performance above 30%. These clusters mostly constitute those in which the original model has non-zero accuracy with an initial performance varying from around 2% to 20%. Then we have one cluster where the accuracy is improved to only ∼10% and three clusters where the model never selects good poses neither before nor after the bootstrapping. These results, further supported by the performance when run using an oracle confidence model in Appendix E.2, suggest that future

---

[2]We release data, instructions, code, and weights at `https://github.com/gcorso/DiffDock`.
[3]Code for these experiments can be found at `https://github.com/LDeng0205/confidence-bootstrapping`.

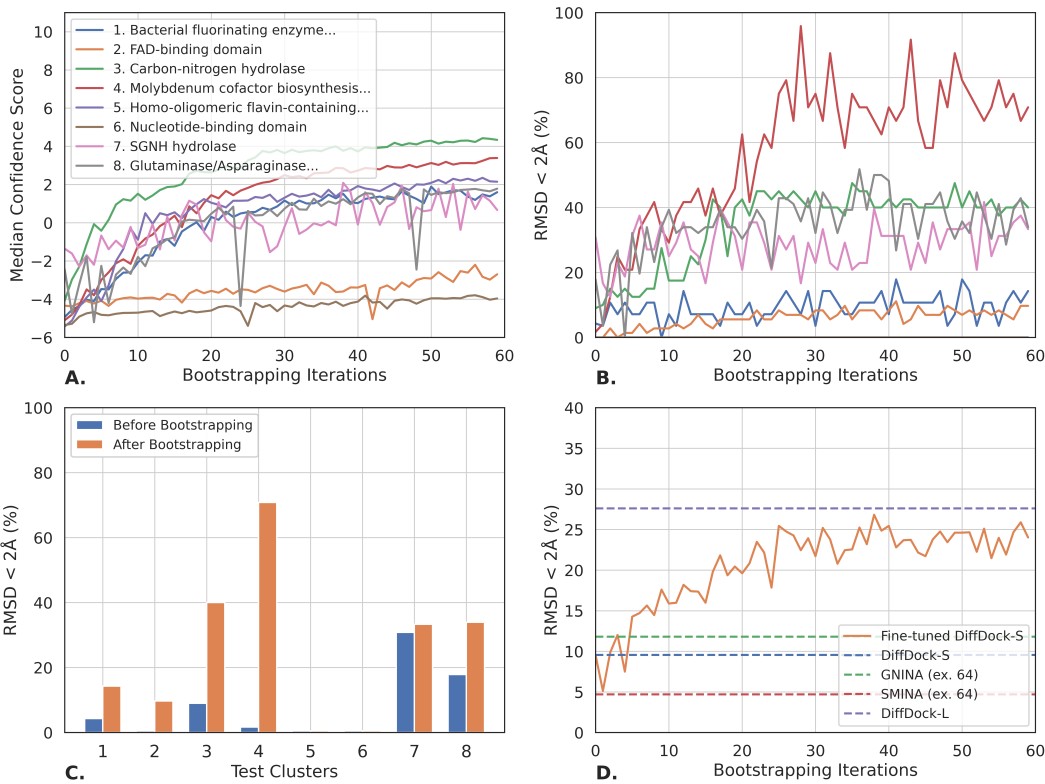

Figure 4: Empirical performance of CONFIDENCE BOOTSTRAPPING across the 8 protein domain clusters within DOCKGEN-cluster. We did two fine-tuning runs for each cluster and report the averaged results. All performances are measured based on top-1 pose when taking 8 samples with the fine-tuned models. **A.** Median confidence of sampled points over at every iteration. **B.** Proportion of top-1 predictions below 2Å over the course of the iterations for each cluster. **C.** Performance for each cluster before the fine-tuning and after the K=60 steps of CONFIDENCE BOOTSTRAPPING. **D.** Aggregated performance along the fine-tuning for all the clusters weighted by their count with, as references, the performance of some of the baselines on the same set.

improvements to either the score or confidence models will lead to even further gains when finetuned with CONFIDENCE BOOTSTRAPPING.

## 6 CONCLUSION

Given the potential utility of high-accuracy blind docking in biology and drug discovery, it is important to track the progress of ML-based methods to generalize to unseen pockets. To this end, we have proposed DOCKGEN, a new benchmark for blind docking generalization based on the classification of binding protein domains. Evaluating existing ML methods on the DOCKGEN benchmark highlights how overfitting training data prevents generalization to unseen binding modes. By scaling the training data and model size as well as integrating a novel synthetic data generation technique, we were able to significantly improve the generalization ability and developed and released DIFFDOCK-L, a new state-of-the-art docking method.

To improve generalization even further, we proposed CONFIDENCE BOOTSTRAPPING, a self-training method that only relies on the interaction between a diffusion and a confidence model and exploits the multi-resolution structure of the sampling process. This allows the direct fine-tuning of the docking model on classes of proteins where binding structural data is not available. Empirically, the method shows significant improvements on the DOCKGEN benchmark, going from 10% to 24% success rate for efficient and fast models. Finally, we believe this opens up the possibility of training even larger-scale docking models that have so far been obstructed by the size and diversity of the available data, bringing us one step closer to a generalizable solution to the docking challenge.

ACKNOWLEDGEMENTS

We thank Bowen Jing, Hannes Stark, Rachel Wu, Peter Holderrieth, Jody Mou, and Saro Passaro for their helpful discussions and feedback. We thank Jacob Silterra for his help in the code deployment.

This work was supported by the NSF Expeditions grant (award 1918839: Collaborative Research: Understanding the World Through Code), the Machine Learning for Pharmaceutical Discovery and Synthesis (MLPDS) consortium, the Abdul Latif Jameel Clinic for Machine Learning in Health, the DTRA Discovery of Medical Countermeasures Against New and Emerging (DOMANE) Threats program, the DARPA Accelerated Molecular Discovery program, the NSF AI Institute CCF-2112665, the NSF Award 2134795, and the GIST-MIT Research Collaboration grant. N.P. acknowledges support from NIH grant R00GM135519 and from the Innovation Research Fund of the Dana-Farber Cancer Institute.

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

## A DockGen Benchmark Details

### A.1 Dataset creation

In this section, we specify the details of how the Binding MOAD dataset was parsed and filtered to obtain the DockGen benchmark validation and test sets. The benchmark was created following these steps:

1. We perform the ECOD-based clustering. Each ligand in a protein is classified by the ECOD domain (t group) of the chain that is making the most contacts with the ligand. The t-group of the ligand is assigned using the consensus of per-residue t-group assignments of each contacting amino acid (4.8Å heavy atom distance) in the dominant chain. We use this approach to classify both the complexes in PDBBind and MOAD. There are 179 clusters in MOAD that do not appear in PDBBind. These 179 are randomly and equally divided between validation and test datasets.

2. Within these clusters, since most docking tools do not support the simultaneous docking of multiple ligands, we discard ligands that are within 4.8Å from the heavy atoms of another ligand.

3. We discard the ligands labeled by MOAD as "part of proteins" as these are constituted mostly by metals and covalent binders.

4. We discard very large ligands formed by more than 60 heavy atoms.

5. Ligands with equal chemical composition that are bound to the same protein are also clustered together forming a single ligand with multiple correct poses.

6. To ensure diversity in the benchmark, if the same ligand appears bound to different biological assemblies of the same PDB entry we only select one of these at random.

7. To avoid overrepresentation of certain ligand classes, we limit the maximum number of ligands to 5 of the same molecule separately in the validation and test sets. This avoids, for example, the presence of a large number of NAD ligands when considering the "NAD-binding domain" cluster.

8. To follow the convention that has been used to train previous ML-based blind docking algorithms, we select only the protein chains that have a C-alpha atom within 10Å of a ligand's heavy atom. In the cases of multiple equal ligands bound to the same complex, one was selected as a reference for the filtering.

After these steps we are left with a validation set composed by 141 unique complexes from 70 different clusters and a test set formed by 189 complexes from 63 clusters.

### A.2 Dataset analysis

In this section, we compare the test set of the newly generated DockGen benchmark and the one of PDBBind.

**Receptor and ligand dimension** In Figure 5 we plot the distribution of the sizes of the receptors and ligands in the two sets. These are not too different, but on average DockGen seems to present slightly smaller ligands but larger protein receptors. The latter might be one of the explanations for why even traditional search-based docking methods that are often considered to generalize relatively well (even though they also present a number of parameters fitted to experimental data) do worse in the DockGen test set.

**Binding site similarity analysis** To evaluate binding site similarity between datasets, we extract residues in contact with the ligand (defined as protein's heavy atoms within 5Å of ligand heavy atoms) and merge them into a single chain to handle binding sites at protein-protein interfaces. We then TMalign every extracted binding site in the training set to every extracted binding site in the test set for both the PDBbind and DockGen datasets. We parse the TMalign output for any aligned residues and record their amino acid identities for both the target and query binding sites. We then compute the fraction of the target and query matched by the alignment and take the harmonic mean of those fractions as a first metric for binding site similarity. We then use the BLOSUM62 amino acid

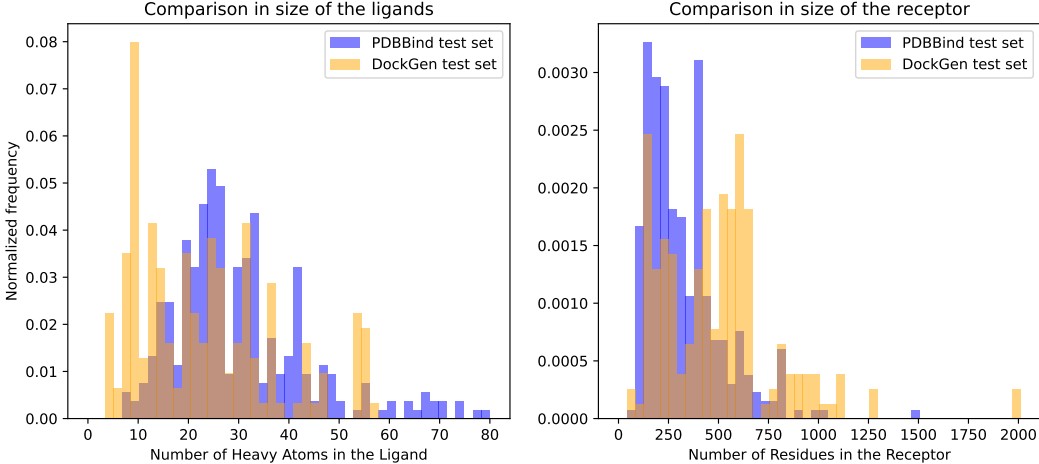

Figure 5: Comparison of the sizes of the ligand and the receptor between the test sets of PDBBind and DOCKGEN.

substitution matrix to quantify similarity in the case that two amino acids are structurally aligned but not exact sequence matches. Amino acids that are more similar have larger values in this matrix. We normalize the BLOSUM matrix to values between 0 and 1 and then take the average of these normalized values across all structurally matched residues as a second similarity metric.

We compare these metrics across the PDBBind and DOCKGEN test sets when compared to PDBBind training set in Figure 2 and show that binding sites in PDBBind tend to be more similar to those in the test set compared to the same comparison for the DOCKGEN dataset: the density in the top right corner indicates significant binding site overlap in PDBBind and little in the case of DOCKGEN.

## B   DATA AUGMENTATION STRATEGIES DETAILS

### B.1   INCREASING THE TRAINING DATA

To increase the training data using the clusters of Binding MOAD that did not go into the validation or test set, we first take the clusters that remained after applying step 1 above. We also remove all complexes resolved in 2019 or later to maintain the validity of the PDBBind test set. Then, we again discard all ligands that are close to other ligands and those with a single heavy atom.

We take the remaining complexes, forming a dataset with 20,012 ligand poses bound to 14,214 different biological units, and use them for training. For ligands that are bound to the same biological unit, at every epoch we select a single ligand at random for training. At training time, we simply concatenate this dataset to PDBBind's training set. One should note that the two datasets are not disjoint and many complexes will appear in both. We avoid removing the redundancy to give more weight to complexes that passed both filtering processes of MOAD and PDBBind and are therefore more likely to be of higher quality.

### B.2   VAN DER MER-INSPIRED TRAINING

To extract van der Mer-like amino-acid ligands, we start from a large collection of protein structures that comprise the ProteinMPNN [Dauparas et al., 2022] training set. At preprocessing time, for every sidechain $s$ in the protein, we compute $n_s$ the number of other amino acids that have heavy atoms within 5Å and are not local in primary sequence (within 10 residues). This determines the extent to which the remaining part of the protein forms a pocket around the selected sidechain, once the adjacent sequence on each side is removed.

At training time, we iterate over the clusters of proteins in the dataset. We use the same clusters adopted by ProteinMPNN that were generated with a sequence similarity cutoff of 30%. For each cluster, we take one protein at random, and within this, we select a sidechain $s$ at random with

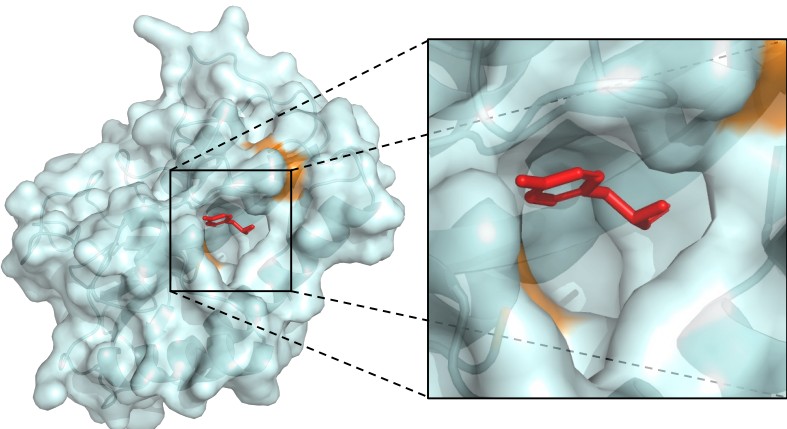

Figure 6: Visualization of the van der Mer-inspired synthetically generated docked poses. In this case, a tyrosine (in red) is taken to be the ligand, and the amino acids that are nearby in the primary sequence are removed from the protein structure of 1QXZ (the created chain breaks are highlighted in orange).

probability proportional to $\max(0, n_s - 5)$ sampling buried residues with a higher likelihood. If no sidechain has more than 5 contacts then the protein is skipped.

Once a sidechain is selected, the sequence-local segment of the protein chain, consisting of 21 residues centered at the selected sidechain (i.e. 10 residues on either side), is removed from the protein structure. The sidechain and its backbone atoms are then used as the ligand that the model is trained to dock. An example of such resulting complex is represented in Figure 6.

Note that for computational efficiency, instead of re-computing the embedding each time we truncate a protein structure, we compute the ESM embeddings [Lin et al., 2022] used by DiffDock only once with the full sequence and then simply crop the removed sequence out (even though the embedding of the remaining residues might be affected by the residues that were removed).

## C    CONFIDENCE BOOTSTRAPPING DETAILS

In this section, we discuss the CONFIDENCE BOOTSTRAPPING procedure and our experimental setup. We demonstrate that CONFIDENCE BOOTSTRAPPING improves blind docking performance of the pretrained score model on a previously unseen cluster without knowing the ground-truth poses of that cluster. This is achieved by iteratively sampling from the score model (the rollout process), evaluating these samples with the confidence model, and updating the score model treating high-quality samples as ground-truth poses. Our procedure assumes a DIFFDOCK score model and a confidence model both pretrained. We use the procedure outlined in Corso et al. [2022] to train and sample from the score model.

### C.1    TRAINING PROCEDURE

The training procedure takes as input ligand/receptor pairs $D = \{d_1, d_2, ...\}$ from a cluster of interest, a pretrained diffusion model parametrized by $\theta$ with learned distribution over poses $p_\theta(\mathbf{x}; d)$ and score $s_\theta(\mathbf{x}^{(t)}, t; d)$, and a pretrained confidence model $c_\phi(\mathbf{x}, d)$. At each step, we sample complexes from the score model through the reverse diffusion process, score these complexes with the confidence model, and use high-confidence samples to update the score model. More specifically, we start with the buffer $B$ that is initially empty; at step $i$, we sample $q$ candidate poses $(\mathbf{x}_j, d_i)$ for $j \in \{1, \cdots, q\}$ for each $d_i \in D$ from $p_{\theta_{ema}^i}(\mathbf{x}; d)$ and evelute these candidate poses with $c_\phi$. We use the exponential moving averaged version of model weights $\theta_{ema}^i$ as in Corso et al. [2022] to stabilize the inference over different training epochs. We add the sampled poses with a confidence score $c_\phi(\mathbf{x}, d) > k$, where $k$ is the confidence cutoff, to the buffer: $B = B \cup \{(\mathbf{x}_j, d_i) \mid c_\phi(\mathbf{x}_j, d_i) > k\}$.

Then, we update the score model $\theta^i$ by treating complexes in $B$ as ground truth poses and performing a fixed number of SGD update steps on the diffusion objective to obtain $\theta^{i+1}$.

To train the original model we use the denoising score matching loss presented in Section 4.1 with a weighting $\lambda(t) = t/\mu_t^2$ where $\mu_t$ is the expected score magnitude at time $t$. During fine-tuning we use $\lambda'(t) = 1/2\mu_t^2$ and $\lambda'(t) = t/\mu_t^2$, to give more emphasis to the bootstrapping examples at high noise levels. Note that these different weightings can also be seen as sampling $t$ from different beta distributions while always normalizing by the expected score.

There are a few nuances to building and using the buffer $B$. First, we resample a constant $m$ complexes from $B$ to train the score model at every step. Motivated by the intuition that higher-score poses should be sampled more frequently, the poses are sampled from the distribution $p(\mathbf{x}, d) \sim \frac{1}{Z} \exp[c_\phi(\mathbf{x}, d)]$ where $Z$ is a normalization constant. Additionally, we enforce an upperbound on the number of samples per complex to encourage diversity across complexes in $B$ by only keeping the $n$ highest scoring poses per protein/ligand pair. Moreover, as alluded to in 4.3, updating the score model with samples from the buffer may cause it to lose information about how to perform steps with $t < t_{\min}$ and/or learn to sample from pathological spaces of the confidence model, where the confidence model assigns high scores to poses distant from the ground truth. Therefore, at each training step, we also include $p$ randomly sampled complexes from MOAD training set in addition to complexes sampled from $B$. In our scheme, $k$, $p$, $q$, $m$, and $n$ are hyperparameters selected through testing the method on the validation dataset.

## C.2 Model Architecture

To enable the efficient execution of the CONFIDENCE BOOTSTRAPPING iterative training routine, we make a number of changes to the architecture of DIFFDOCK's score (obtaining what we refer to as DIFFDOCK-S) and confidence models.

**Score model**  We speed up the architecture and execution of DIFFDOCK's score model in a number of ways. First, we add a number of embedding message-passing layers which, unlike the cross-attentional interaction layers of DIFFDOCK, independently process the protein and ligand structures. This allows us to increase the depth of the architecture with very little added runtime. In fact, due to the significantly higher number of nodes, the main complexity of the embedding layer lies in the protein component. However, under the rigid protein assumption, the structure is the same across all the different samples and diffusion steps. Therefore, the protein embedding can be computed only once resulting in minor computational overhead. Further, we limit the order of the spherical harmonics used to represent the edges to one (with minor loss in accuracy), and we batch and move the whole reverse diffusion process to GPU to improve parallelization and memory transfers.

**Confidence model**  We also change a number of design choices in the confidence model to make it more efficient and better suited for CONFIDENCE BOOTSTRAPPING. The first choice is to force the model to reason about local interactions, as the affinity of a ligand to a particular pose is largely a local property. We achieve this by simply feeding to the model the receptor structures of only the amino acids whose C-alpha is within 20Å of any of the predicted ligand atoms' positions.

Further, during the binary classification training of the confidence model, we try to remove the bias of the model against hard targets by balancing the proportion of positive and negative examples the model is trained over. When not possible with sampled poses (because the model only samples negatives), we use the so-called conformer-matched ligand poses [Jing et al., 2022] as positive examples. Moreover, to make the transition smoother and reduce the perceived variance in the labels, we separate the positive (poses with RMSD<2Å) and negative classes (>4Å)[4]. Finally, we supervise the model not only to predict the label of the whole ligand but also that of each individual atom (supervised on their individual distance to the ground truth pose).

## C.3 Experimental Details

**Setup**. We tested CONFIDENCE BOOTSTRAPPING on DOCKGEN-clusters, a subset of DOCKGEN containing 8 clusters with at least 6 complexes each. For every cluster in DOCKGEN-clusters, we

---

[4]We note that concurrent work Masters et al. [2023] also applies this strategy.

started with the same pretrained diffusion and confidence models, and ran 60 iterations of CONFI-DENCE BOOTSTRAPPING, where each iteration contained a rollout step and an update step with 200 SGD updates. We did two evaluation runs with the same hyperparameters and reported the averaged results. To evaluate the generated complexes, we computed the symmetric-corrected RMSD of sPyRMSD [Meli & Biggin, 2020] between the predicted and the crystal ligand atoms. We reported the percentage of top-ranked predictions that have an RMSD less than 2Å, a metric generally considered to be representative of getting the correct pose [Alhossary et al., 2015; Hassan et al., 2017; McNutt et al., 2021; Corso et al., 2022].

**Hyperparameters**. In our experiments, we chose confidence cutoff $k = -4$, number of complexes sampled from PDBBind $p = 100$, number of complexes sampled from the buffer $m = 100$, number of inference samples $q = 32$, and maximum samples per protein/ligand pair $n = 20$. At the rollout step, we ran 4 inference steps each with 8 samples and compute the RMSD less than 2Å metric with the top-ranked pose from each inference step to reduce variance in the reported metric. Additionally, we set number of inference samples to 80 for the first bootstrapping inference step to fill the initially empty buffer. These parameters were selected by testing the method on the 5 DOCKGEN validation clusters.

## C.4 COMPUTATIONAL COST

The runtime cost of our fine-tuning approach depends on the number of complexes sampled and the number of gradient update steps. In our experiment, on average we sample 320 complexes in the cluster for every 200 gradient update steps (with batch size of 5). We interleave these two operations 60 times. With these parameters and training on one NVIDIA A6000 GPU, the average run time is 8 hours.

One can try compare this cost with for example running GNINA (here we assume the default version without extra search exhaustiveness), which generalizes better without retraining but it is considerably slower and cannot be finetuned on specific domains. Assuming the methods (GNINA and DiffDock w/ confidence bootstrapping) work similarly, then one can ask in which setting would it be faster to run either one of them. It takes 232 inference complexes for DiffDock to ammortize the cost of the finetuning and from then onwards it can provide significant runtime improvements. As many screening campaigns require significantly more than 232 complexes, we believe that the idea of finetuning a docking model to the specific target class of interest may prove a useful feature in the future.

## D   CONFIDENCE BOOTSTRAPPING AND REINFORCEMENT LEARNING

In the section we discuss the connection of CONFIDENCE BOOTSTRAPPING to Reinforcement Learning (RL). Our objective can be approximately formulated as finding some score model parameters $\theta^*$ to maximize the confidence conditioned on a cluster, and can be written as

$$\theta^* = \underset{\theta}{\operatorname{argmax}} \, J_\theta$$

where

$$J_\theta := \mathbb{E}_{p_\theta(\mathbf{x};d)}[c_\phi(\mathbf{x};d)]. \tag{1}$$

This objective can be directly formulated as the RL objective, where the reward function is the confidence model $c_\phi$ and the policy is the diffusion model $p_\theta(\mathbf{x};d)$. Then, our method can be seen as loosely related to the policy gradient method, where the gradient of Equation 1 can be written as

$$\begin{aligned}
\nabla_\theta J_\theta &= \nabla_\theta \mathbb{E}_{p_\theta(\mathbf{x};d)}[c_\phi(\mathbf{x};d)] \\
&= \mathbb{E}_{p_\theta(\mathbf{x};d)}[\nabla_\theta \log p_\theta(\mathbf{x};d) c_\phi(\mathbf{x};d)] \\
&\approx \frac{1}{n} \sum_{i=1}^n \nabla_\theta \log p_\theta(\mathbf{x}_i;d) c_\phi(\mathbf{x}_i;d)
\end{aligned}$$

In the policy gradient method, $n$ samples are collected to approximate the expected value, which corresponds to the number of poses we sample per complex during training. However, computing $\nabla \log p_\theta(\mathbf{x}_i;d)$ is intractable for diffusion models. Instead, we modify the reward to $\exp(c_\phi(\mathbf{x};d))$ and the objective to:

$$\theta* = \underset{\theta}{\arg\max} \ \mathbb{E}_{p_\theta(\mathbf{x};d)}[\exp(c_\phi(\mathbf{x};d)) \log p_\theta(\mathbf{x})]$$
$$= \underset{\theta}{\arg\min} \ \mathbb{E}_{p_\theta(\mathbf{x};d) \exp(c_\phi(\mathbf{x};d))}[-\log p_\theta(\mathbf{x})]$$
$$= \underset{\theta}{\arg\min} \ \mathbb{E}_{p_{\theta^i,\phi}(\mathbf{x},d)}[-\log p_\theta(\mathbf{x})]$$

Again this cannot be estimated directly but we can use the denoising score matching loss with $p_{\theta^i,\phi}(\mathbf{x}, d)$ as our data distribution to derive an upper bound that can be optimized [Song & Ermon, 2019]:

$$\mathbb{E}_{p_{\theta^i,\phi}(\mathbf{x},d)}[-\log p_\theta(\mathbf{x})] \leq \mathcal{J}_{DSM}(\theta) + C$$

where $C$ is a constant that does not depend on $\theta$ and $\mathcal{J}_{DSM}(\theta)$ is the denoising score matching loss.

**Avoiding overoptimization**    Note that unlike traditional RL problems we not only have the (denoising score matching) loss based on the *reward* from our confidence model, but we also keep the loss component that came from our training set optimization. While the RL objective is to directly maximize reward/confidence, we also want to prevent the model from overfitting into pathological spaces of the confidence model (where the confidence model gives high confidence for bad poses), aka "overoptimization", which is why we included the real samples from training set.

**RL in docking**    Reinforcement learning has been applied in the past for the docking task, for example, Wang et al. [2022] uses it for protein-ligand docking while Aderinwale et al. [2022] applies it to multimeric protein docking. These works have two strong differences with our approach: (1) RL is used to optimize the pose at inference time for a specific complex not to train/fine-tune a model (2) the specific methodologies used are very different from the idea of CONFIDENCE BOOTSTRAPPING.

# E    ADDITIONAL EXPERIMENTS

## E.1    POSEBUSTERS BENCHMARK

We report in Table 2 the results of DIFFDOCK-L on the PoseBusters benchmark test set compared to the recent ROSETTAFOLD-ALLATOM [Krishna et al., 2023] and all the methods reported by Buttenschoen et al. [2024]: GOLD [Jones et al., 1997], VINA [Forli et al., 2016], DEEPDOCK [Méndez-Lucio et al., 2021], UNI-MOL [Zhou et al., 2023], EQUIBIND [Stärk et al., 2022], TANKBIND [Lu et al., 2022] and DIFFDOCK [Corso et al., 2022]. Even on this benchmark DIFFDOCK-L performs best among blind docking methods.

## E.2    CONFIDENCE BOOTSTRAPPING ORACLE EXPERIMENTS

In a separate experiment, we tested CONFIDENCE BOOTSTRAPPING with "oracle" confidence predictions. The oracle confidence predictor is a monotonic transformation of the true RMSD between the ground truth pose and the predicted pose: $c_\phi(\mathbf{x}, d) = -4 \tanh\left(\frac{2}{3}RMSD(\mathbf{x}, \mathbf{x}^*) - 2\right)$. Note that this is not a practical setting as we would not have access to the ground truth poses $\mathbf{x}^*$ in real applications. However, this is an illustrative experiment establishing an upperbound on the performance gains achievable through CONFIDENCE BOOTSTRAPPING with a "perfect" confidence model (Figure 7).

## E.3    CONFIDENCE BOOTSTRAPPING ABLATION STUDIES

To better understand the benefits of different components of CONFIDENCE BOOTSTRAPPING, we conducted ablation experiments and report the ablation results (Figure 8) on the DOCKGEN validation dataset. This validation dataset is selected by choosing all clusters with four or more complexes in DOCKGEN, excluding those in DOCKGEN-clusters. We further exclude one cluster where DIFFDOCK-L already achieves higher than $80\%$ RMSD less than 2, and another where an insufficient number of samples pass the confidence threshold, resulting in 7 clusters. Here, we provide a description of each ablation experiment.

Table 2: Comparison of the performance on the PoseBusters dataset. Pocket-based docking methods receive further information about the location and shape of the binding pose. [†] unlike the other blind docking methods reported ROSETTAFOLD-ALLATOM (1) does not take as input the holo-structure of the protein, (2) does take as input the information about the cofactors involved in the binding, and (3) uses as training cutoff date April 2020 instead of December 2018.

| Method | RMSD $\leq$ 2Å |
|---|---|
| **Pocket-based docking** | |
| GOLD | 58% |
| VINA | 60% |
| DEEPDOCK | 20% |
| UNI-MOL | 22% |
| **Blind docking** | |
| EQUIBIND | 2% |
| TANKBIND | 16% |
| DIFFDOCK | 38% |
| ROSETTAFOLD-ALLATOM[†] | 42% |
| DIFFDOCK-L | 50% |

**Temperature = 0**  Instead of sampling from the buffer with $p \propto \exp[c_\phi(\mathbf{x}, d)]$, which can be interpreted as $p \propto \exp[k c_\phi(\mathbf{x}, d)]$ with temperature $k = 1$, we investigate the effects of drawing buffer samples with uniform probability ($t = 0$).

$t$ **sampling**  We change the distribution of $t \in [0, 1]$, the amount of noise added to each sample during training time. Heuristically, we would prefer to sample higher values of $t$ more often to finetune the earlier stages of the reverse diffusion process, guiding the model into finding the right pocket. $t$ is sampled from a beta distribution $\text{Beta}(\alpha, \beta)$. In addition, samples from $p_{\text{train}}$ use a different value of $\alpha$ and $\beta$ from buffer samples. In CONFIDENCE BOOTSTRAPPING, we have $\alpha = 2$, $\beta = 1$ for buffer samples, and $\alpha = 1$, $\beta = 1$ for $p_{\text{train}}$ samples, with $t_{\min} = 0$. We experiment with different variations of these default parameters (Figure 8).

**Removing samples from $p_{\text{train}}$**  In this experiment, we remove $p_{\text{train}}$ complexes (real samples) from the update steps of the diffusion model, using only samples from the buffer. These complexes were initially added to avoid the overoptimization problem as discussed previously, and removing them, unsurprisingly, decreases the performance of our method (though our method still significantly outperforms the baseline DiffDock).

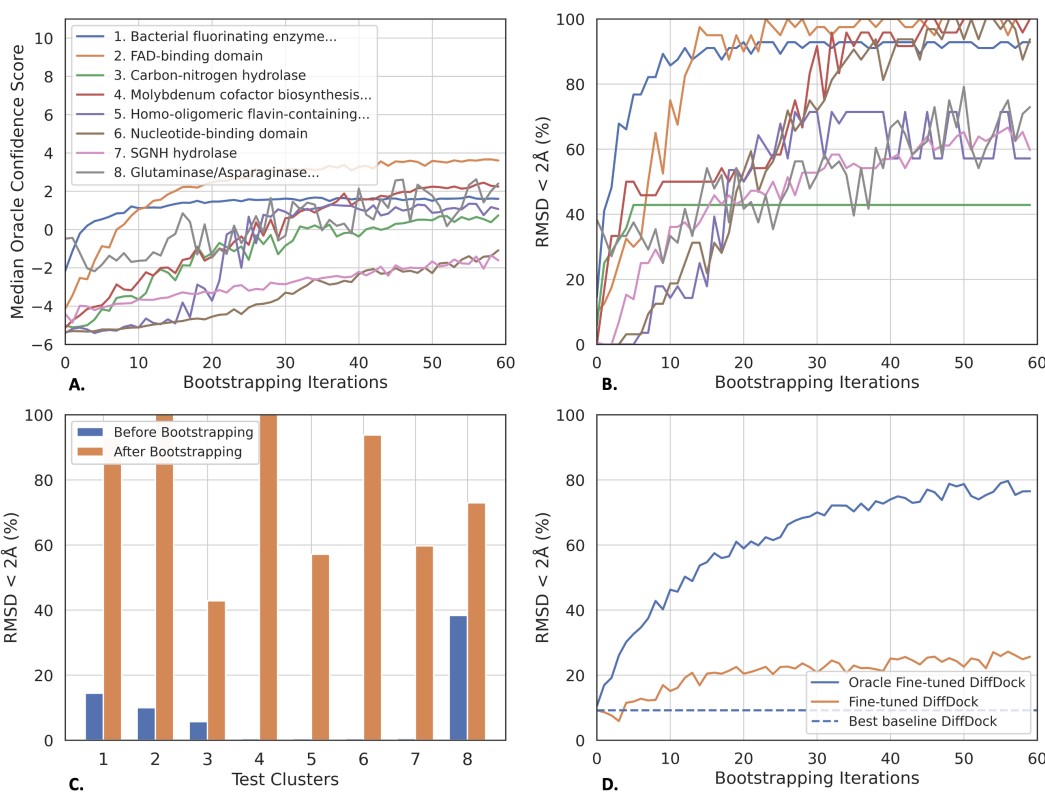

Figure 7: Empirical performance of CONFIDENCE BOOTSTRAPPING with oracle confidence across the 8 protein domain clusters within DOCKGEN-cluster, the bootstrapping method is run twice for every cluster and we show the average results of the two runs. All performances are measured based on the top-1 pose when taking 8 inference samples with the fine-tuned models. **A.** Median confidences of sampled points at every iteration for each cluster. **B.** Proportion of top-1 predictions below 2Å over the course of the iterations for each cluster. **C.** Performance for each cluster before the fine-tuning and after the K=60 steps of CONFIDENCE BOOTSTRAPPING. **D.** Aggregated performance for all the clusters weighted by their number of complexes, showing results using the oracle confidence model and pretrained confidence model.

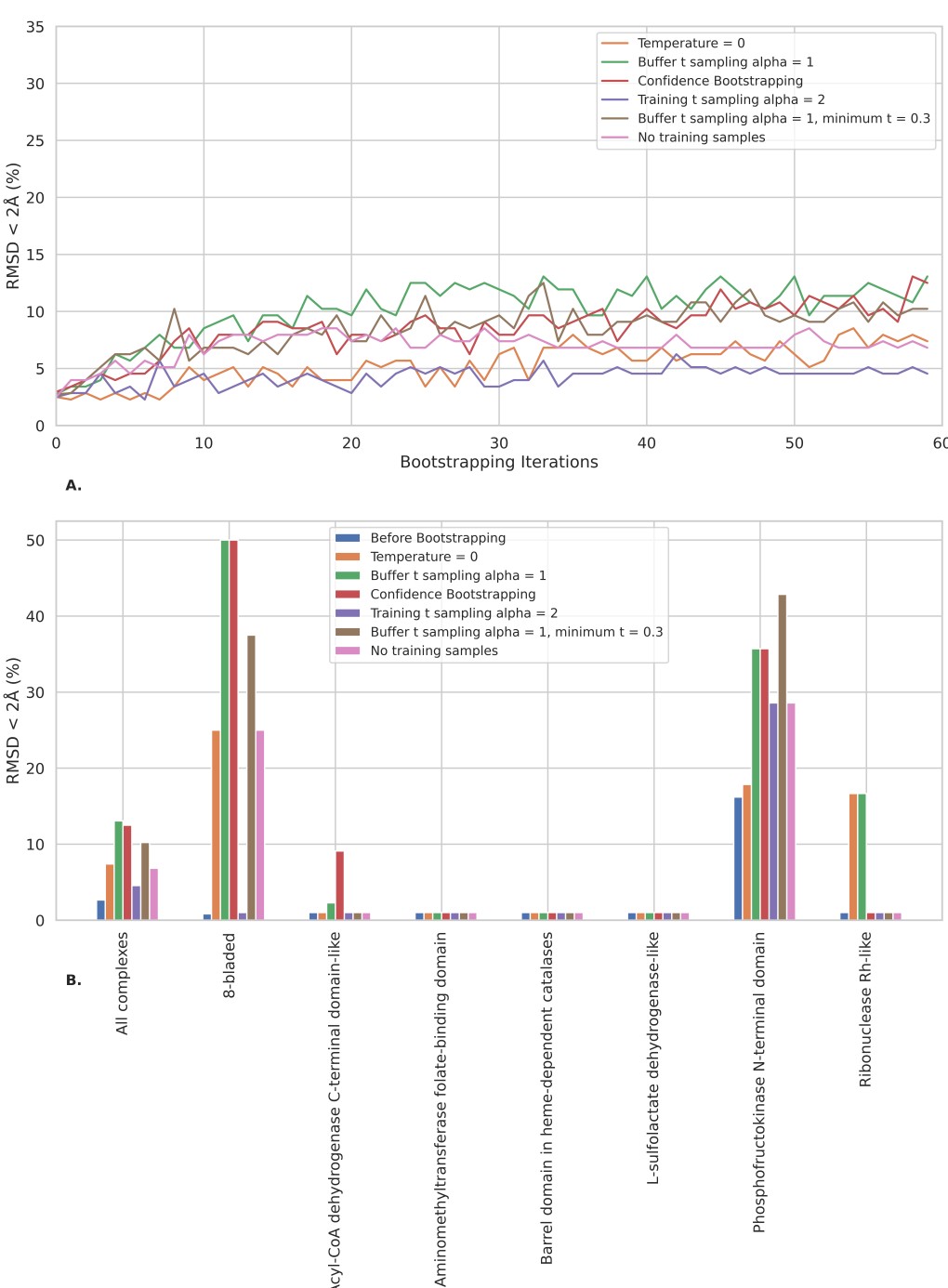

Figure 8: Ablation studies on CONFIDENCE BOOTSTRAPPING tested on the 7 protein domain clusters within the validation set. All performances are measured based on the top-1 pose when taking 8 inference samples with the fine-tuned models. No Real Samples refer to the algorithm using only buffer samples in the loss, and Alpha=3 refers to the altered distribution of noise schedule during training. **A.** Aggregated performance for all the clusters weighted by their number of complexes, showing results for the different ablation experiments. **B.** Performance for each cluster before the fine-tuning and after the K=60 steps of CONFIDENCE BOOTSTRAPPING.

