# OpenReview forum: "Deep Confident Steps to New Pockets: Strategies for Docking Generalization"
_ICLR.cc/2024/Conference — ICLR 2024 poster_

### Official Review · Reviewer_Jhgj · 2023-10-28

**Soundness:** 3 good
**Presentation:** 4 excellent
**Contribution:** 3 good
**Rating:** 6
**Confidence:** 2

**Summary:**

A new benchmark and a novel training paradigm have been developed to enhance generalization capabilities. The effectiveness of the model has been demonstrated through experiments.

**Strengths:**

The experimental section is comprehensive and well-designed. The proposed model demonstrates good performance on specific tasks.

**Weaknesses:**

Due to my current level of expertise, I have not identified any shortcomings in this paper.

**Questions:**

I don't have any further questions at the moment.

---

> ### Author Response · Authors · 2023-11-18
> **Response to Reviewer Jhgj**
>
> Thank you very much for taking the time to review the paper and for your positive feedback. We invite you to check our updated manuscript as we have made a number of improvements. We hope these may help strengthen your confidence in the contribution of our work. We are happy to answer any question or integrate any suggestion you may have.

---

### Official Review · Reviewer_Eb79 · 2023-11-02

**Soundness:** 3 good
**Presentation:** 2 fair
**Contribution:** 2 fair
**Rating:** 5
**Confidence:** 4

**Summary:**

The paper proposed a new benchmark in the field of docking. Under more strict data split, the authors claimed that currently developed deep learning methods failded to generalize to unseen proteins. A series of efforts were made to tackle such limitations, including architecture modifications, increasing the training data, Van der Mer-inspired docking and most important confidence bootstrapping. However, despite of a brilliant research idea and meaningful topic, the paper failed to justify their contributions with convincing and reproduciable experiments.

**Strengths:**

- The DockGen Benchmark provides a new perspective to the evaluation of ML docking methods. Taking the similarity of binding pockets into account is more reasonable.
- Confidence bootstrapping provides a new data-free methodology to improve docking models on unseen proteins.

**Weaknesses:**

- The paper mentioned a lot of works in terms of neural network architecture and data augmentation. However, I could not see their connections to the main topic of generalization. In Table 1, all those attempts failed to improve DIFFDOCK performance in the DockGen benchmark in terms of success rate(<2A).
- Although a code repo is provided, I can not find any details about how baselines are benchmarked with the DockGen dataset. Specifically, when new data-split is provided, it is important to report whether baselines are re-trained or not. I am also confused by unclearified marks like EX. 64
- The effect of confidence bootstraping is not approperately evaluated. Although the performance is demonstrated on 8 clusters, it is documented that there are 63 clusters in the test set. It is not convincing to claim the proposed method is effective before a more systematic and quantitative evaluation.

**Questions:**

- Why is ECOD used for data split? Best to my knowledge, SCOP(e) and CATH are more widely acknowledged.
- 1QXZ and 5M4Q have a sequence similarity of 22% but not 16%. I blast it myself.
- In Table 1, do you retrain GNINA or other machine learning methods under the DockGen split?
- Why do traditional methods like SMINA not perform well on the DockGen benchmark?
- Only 4 out of 8 clusters are proved to be improved by confidence bootstraping. Why are other clusters not improved?
- In the paper Do Deep Learning Models Really Outperform Traditional Approaches in Molecular Docking? (https://arxiv.org/pdf/2302.07134.pdf), a new baseline of DIFFDOCK+Vina(UniDock) is proposed. It would be more approperate to be compared with than P2RANK+SMINA.

---

> ### Author Response · Authors · 2023-11-18
> **Response to Reviewer Eb79 - Part 1/2**
>
> Thank you very much for your feedback and questions, we truly appreciate the effort and time you spent on the review. We responded to each of your comments and questions below and we adapted the manuscript to reflect these changes.
>
> **Q: The paper mentioned a lot of works in terms of neural network architecture and data augmentation. However, I could not see their connections to the main topic of generalization. In Table 1, all those attempts failed to improve DIFFDOCK performance in the DockGen benchmark in terms of success rate(<2A).**
>
> The reason why we tested these different approaches is that a few of the most widely used techniques to improve generalization used with neural networks are (1) improve inductive biases in neural architecture, (2) get more data, (3) perform data augmentation. Indeed, these strategies do not seem to provide significant improvements in this setting (with arguably the exception of data increase). We do report these "failed attempts" because we believe it gives the reader a better understanding of the tradeoff between different approaches. We clarified this point in the manuscript.
>
> **Q: Although a code repo is provided, I can not find any details about how baselines are benchmarked with the DockGen dataset. Specifically, when new data-split is provided, it is important to report whether baselines are re-trained or not. I am also confused by unclearified marks like EX. 64… In Table 1, do you retrain GNINA or other machine learning methods under the DockGen split?**
>
> Thank you for raising these points. We tried to clarify them below and in the manuscript. The DockGen benchmark was designed specifically to be compatible with the dataset and splits the baselines were originally trained on i.e. the (validation and) test sets contain complexes that belong to different clusters from those seen in PDBBind's training and validation set. Therefore none of the baselines required retraining. Ex. indicates the level of search exhaustiveness that can be regulated in search-based methods.
>
> **Q: The effect of confidence bootstrapping is not approperately evaluated. Although the performance is demonstrated on 8 clusters, it is documented that there are 63 clusters in the test set. It is not convincing to claim the proposed method is effective before a more systematic and quantitative evaluation.**
>
> As a reminder, the model is finetuned on individual protein clusters. We have selected the 8 clusters in the test set by including all clusters that have 6 or more complexes. Most clusters contain 2 or less complexes, and it is difficult to fine-tune on clusters with very little structural or binding data (on top of being computationally very expensive to finetune 63 different models). Although we only include 8 clusters out of 63 in the test set, the 8 clusters actually contain 85 out of 189 complexes in the test split. We clarified these points in the manuscript.
>
> **Q: Why is ECOD used for data split? Best to my knowledge, SCOP(e) and CATH are more widely acknowledged.**
>
> We largely expect ECOD, SCOP(E), and CATH to give similar splits (see the introduction of [1]). We used ECOD also because it is automated and applied to every PDB entry.
>
> [1] R. D. Schaeffer, J. Zhang, L. N. Kinch, J. Pei, Q. Cong, N. V. Grishin, Classification of domains in predicted structures of the human proteome. Proc. Natl. Acad. Sci. U. S. A. 120, e2214069120 (2023)
>
> **Q: 1QXZ and 5M4Q have a sequence similarity of 22% but not 16%. I blast it myself.**
>
> Thank you for pointing this out. We obtained the 16% value with the pw2.align.globalxx routine of biopython. However, we recognize that BLAST is a more widely accepted score and have updated the manuscript accordingly.
>
> **Q: Why do traditional methods like SMINA not perform well on the DockGen benchmark?**
>
> We believe it is for a combination of reasons: (1) they also contain some parameters that have been fitted to protein domain clusters that are unlikely to be contained in our benchmark (i.e. the receptors are harder than for example those in PDBBind), (2) the receptors present in our test set are on average larger than those present in PDBBind's test set making the docking prediction potentially harder. We now plot these statistics and discuss this point in Appendix A.2.

---

> ### Author Response · Authors · 2023-11-18
> **Response to Reviewer Eb79 - Part 2/2**
>
> **Q: Only 4 out of 8 clusters are proved to be improved by confidence bootstraping. Why are other clusters not improved?**
>
> Indeed of out the 8 clusters we see 4 with significant improvements, 2 with marginal improvements and 2 with no improvements. The level of success of confidence bootstrapping seems to be largely dependent on the initial quality of the score model in the given cluster: if no good poses are ever sampled by the model the bootstrapping process has no way of succeeding. In other words, for the bootstrapping process to succeed the initial generative model needs to have some (non-negligible) coverage of the correct poses in its distribution. One possible strategy to alleviate this would be to use a combination of confidence models trained to recognize poses at larger cutoffs (current one was trained with 2A), as discussed in the paper, the larger the cutoff the less likely will be the confidence model to generalize. We demonstrate this point in Figure C.1 where, with an oracle confidence model that gives signal in the right direction even beyond 2A, the model is able to improve significantly across all clusters.
>
> **Q: In the paper Do Deep Learning Models Really Outperform Traditional Approaches in Molecular Docking? (https://arxiv.org/pdf/2302.07134.pdf), a new baseline of DIFFDOCK+Vina(UniDock) is proposed. It would be more approperate to be compared with than P2RANK+SMINA.**
>
> Thank you for the suggestion, we have added the baseline DiffDock+GNINA (an improved version of Vina) to Table 1.
>
> ___
>
> Thank you again for the very constructive feedback! We are happy to integrate any further suggestion you may have and we hope the improvements have helped increase your conviction on the value of our work.

---

### Official Review · Reviewer_f6Xk · 2023-11-02

**Soundness:** 3 good
**Presentation:** 3 good
**Contribution:** 3 good
**Rating:** 5
**Confidence:** 5

**Summary:**

This paper proposes a new benchmark and training method for evaluating and improving the generalization ability of machine learning models for molecular docking. The key points are:

- Existing docking benchmarks are limited in diversity, focusing on common binding modes. A new benchmark DOCKGEN is proposed based on binding protein domains to better test generalization.
- ML docking models show very weak generalization on DOCKGEN. Increasing training data and data augmentation do not help much.
A new training approach "Confidence Bootstrapping" is proposed. It trains a diffusion docking model using high-confidence poses judged by a separate confidence model, without ground truth data.
- Confidence Bootstrapping significantly improves model accuracy on DOCKGEN, doubling search-based methods. It is more sample-efficient than self-training.

**Strengths:**

- DOCKGEN is a valuable new benchmark to rigorously test model generalization in docking. I really like this work about this kind of new evaluation of the current dataset problem and create a new split setting.
- Thorough experiments demonstrate current ML methods generalize poorly, motivating new approaches.
- Confidence Bootstrapping is innovative, exploiting diffusion model structure and confidence feedback.
- Strong gains shown on DOCKGEN highlight its potential for advancing blind docking.

**Weaknesses:**

- Confidence boostrapping is much like a RL-based method, therefore the authors need to add more discussions about RL and its related works in docking and science.
- There is one recent work that achieves strong performance on unseen targets (though split by UniProt ID), the authors are strongly encouraged to compared the performance on that model: FABind, https://arxiv.org/abs/2310.06763
- More analysis needed on what protein features DOCKGEN probes that past benchmarks missed.
- Ablation studies could better isolate benefits of different components of Confidence Bootstrapping.
- Computational cost and sample efficiency could be compared to alternate training schemes.

**Questions:**

NA.

---

> ### Author Response · Authors · 2023-11-18
> **Response to Reviewer f6Xk - Part 1/2**
>
> Thank you very much for your time and effort spent in reviewing and for the constructive feedback. We integrated all your suggestions in the updated version of the manuscript and summarized the additions below.
>
> **Q: Confidence boostrapping is much like a RL-based method, therefore the authors need to add more discussions about RL and its related works in docking and science.**
>
> Thank you for raising this point. We have updated the manuscript to include a discussion on the connection between RL and our method. This is presented in detail in Appendix D, where we present the mathematical and practical similarities and differences of our approach with policy gradients. In brief, the direct application of the approach of policy gradient to diffusion models fails due to the intractability of the likelihood function, however, we show that by modifying the RL objective, Confidence Bootstrapping can be seen as maximizing a lower bound to the expected objective (we leave the specific details to Appendix D). Further, we discuss the relation of our work with previous application of RL for protein-protein or protein-ligand docking.
>
> **Q: There is one recent work that achieves strong performance on unseen targets (though split by UniProt ID), the authors are strongly encouraged to compared the performance on that model: FABind, https://arxiv.org/abs/2310.06763**
>
> Thank you for the suggestion. We tried running FABind on the DockGen benchmark but have so far been unsuccessful. The repository does not give precise instructions on how to preprocess the data (in README the inference on custom complexes section is referred as "coming soon"). It does mention that the preprocessing is the same as TANKBind, but when we tried using the preprocessed data we generated for TANKBind we ran into some errors. We will reach out to the authors to understand how to run the model and will update the results in the future.
>
> **Q: More analysis needed on what protein features DOCKGEN probes that past benchmarks missed.**
>
> Although it is hard to point out specific protein features that are captured in DockGen but not in PDBBind's test set, the main property that DockGen is able to probe is the capacity of generalization to new binding modes. To highlight this we conduct a careful analysis of the binding sites in the various datasets in Appendix A.2 comparing the similarities that the different benchmarks have with the training set.
>
> In particular, we extract the binding sites of different complexes and then compare them both at a sequence/chemical (using the BLOSUM62 amino acid substitution matrix to quantify the similarity of aligned residues) and structural (using the TMalign output for any aligned residues) level. The results, shown in Figure 5, show significant overlap in the pockets in PDBBind and very little similarity when looking at DockGen pockets. More details on this analysis can be found in Appendix A.2.
>
> **Q: Ablation studies could better isolate benefits of different components of Confidence Bootstrapping.**
>
> Thank you for raising this suggestion. We have now included a number of the ablation results of different components of the method (use of buffer, buffer size, use of training examples, sampling schedule…) in Appendix E.

---

> ### Author Response · Authors · 2023-11-18
> **Response to Reviewer f6Xk - Part 2/2**
>
> **Q: Computational cost and sample efficiency could be compared to alternate training schemes.**
>
> Thank you for raising this point. We have now added a discussion on the computational cost in Appendix C.3 and refer to it in the main text as well. The runtime of our approach depends on the number of complexes sampled and the number of gradient update steps. In our experiment, on average we sample 320 complexes in the cluster for every 200 gradient update steps (with a batch size of 5). We interleave these two operations 60 times. With these parameters and training on one Nvidia A6000 GPU, the average run time is 8 hours.
>
> Unfortunately, it is hard to evaluate the sample efficiency of our approach with radically different training schemes, as no such method for finetuning has been proposed before. However, in the ablation experiments of Appendix E one can look at the rate with which the performance of the finetuned models improves to determine the relative sample efficiency of different variations of the method.
>
> One can also try to compare this cost with for example running GNINA, which generalizes better without retraining but it is considerably slower and cannot be finetuned on specific domains. Assuming the methods (GNINA & DiffDock w/ confidence bootstrapping) work similarly, then one can ask in which setting would it be faster to run either one of them. It takes 232 inference complexes for DiffDock to amortize the cost of the finetuning and from then onwards it can provide significant runtime improvements. As many screening campaigns require significantly more than 232 complexes, we believe that the idea of finetuning a docking model to the specific target class of interest may prove a useful feature in the future.
>
> ___
>
> Thank you again for the very constructive feedback! We are happy to integrate any further suggestion you may have and we hope the improvements have helped increase your conviction on the value of our work.

---

### Official Review · Reviewer_kTFp · 2023-11-02

**Soundness:** 4 excellent
**Presentation:** 4 excellent
**Contribution:** 4 excellent
**Rating:** 8
**Confidence:** 3

**Summary:**

The authors address the general question of assessing and improving generalization of blind docking.
Their contribution are two-fold.
First, they establish a new benchmark for blind docking, coined DockGen, aiming at assessing better the generalization capabilities of docking methods to unseen binding pockets.
This benchmark is mostly built upon building new validation and test sets, composed of complexes with binding pockets that are not present in the "classic" training set of PDBBind.
Using this new benchmark, they show that there exist an important generalization that existing methods, especially ML-based ones, fail at bridging (and which was largely invisible with the classic PDBBind benchmark).
Second, after showing that classic data augmentations strategies fail at bridging this gap, they suggest a new training paradigm, coined Confidence Bootstrapping, that aims at leveraging the feedback from a confidence model to guide the diffusion of a pre-existing (although modified for this purpose) method called DiffDock.
After justifying the approach and formalizing it, the authors benchmark their bootstrapping method, showing great improvement over the already published DiffDock it is based on (and other methods).

**Strengths:**

The first contribution itself (DockGen) is already very valuable in itself as it sheds light on some of the shortcomings of current ML-based docking methods.
While it is very common that widely used benchmarks in the biomedical field fail at capturing the real (and meaningful) difficulty of the task at hand, it is always a great contribution when this comes to light and when a novel benchmark is suggested to address it.
The use of the ECOD classification to cluster complexes in a more "meaningful way" than the more traditional chronological or sequence similarity -based approach is well motivated and using separate external complexes from MOAD allows to maintain the training set of PDBBind that everyone has been using in the field, making it easy for researchers of the field to use DockGen to take a deeper look on the generalization properties of their method.
The results coming from that benchmark give interesting insights on further limitations of current ML-based docking methods.
Despite the very positive results that Confidence Bootstrapping obtains on that benchmark, it does not give (by any mean) the impression that it has has been established for the sole reason of making their new method "look good" (which can sometimes happen).

The second contribution is also well-motivated by well-known yet crucial observations on the nature of the docking task (i.e. it is hard to generate a good pose but "easy" to check if a pose is good).
Before jumping into its formalization, the authors do a very good job at explaining their approach on a higher level, risking some insightful analogies to seemingly loosely related approaches (e.g. AlphaGo), making the reading of the paper a very pleasant experience.
Despite the importance of the contributions and the very limited length of the appendices, the authors also make a good job at giving enough details to extensively describe their approach and allow to reproduce the results.

Last but not least, the results obtained by the new bootstrapping strategy seem to produce very encouraging results in terms of generalization to unseen clusters, although it is still completely failing at generalizing to some of the clusters.
In fact, the results seem so good that they may offer a proper, practical alternative to search-based approaches.

**Weaknesses:**

One main concern I have is related to the relationship of the current manuscript with the already published DiffDock.
While the Confidence Bootstrapping is very specific to diffusion models and clearly builds on top of DiffDock, the authors have made a number of somewhat significant changes to it, even before testing the Confidence Bootstrapping (i.e. the results in Table 1).
I think it makes things a bit confusing for readers that are already very familiar with the field and the method (which I was not).
On a side note, while I do not think that the anonymity of this work has been technically breached, I think that this lack of clear separation between (Corso et al. 22) and the current manuscript gives hints that they are likely to have been authored by the same people.

Although the writing and presentation is overall very good, I find a few small parts to lack clarity.
For instance in Section 4.2 (paragraph 4), it is not really clear to me why the more local and easier task of testing the goodness of a pose can make it easier to generalize to unseen targets (although the experimental results sure seem to suggest so).
The relationship to the multi-resolution structure of diffusion models is also not very clear to me.
I understand that the manuscript is already dense, but in Section 4.4 the explanation and justification on the choice to limit the order of the spherical harmonics is a bit cryptic, especially compared to the level of details of the rest.

To get a clearer overall picture of the performance of the proposed approach(es), I think it would make sense to add the Confidence Bootstrapping results in Table 1 (even if results would only be reported in the DockGen-clusters column).
In addition to be able to see more clearly the improvement in terms of performance, it would also be useful to see the runtime of this approach.
I have no insight on how costly the confidence bootstrapping is, from a runtime perspective.

**Questions:**

In addition to comments / answers to the points I raised in the weaknesses section, I have very few extra additional question.
The choice, during the update step, of using a ratio of half-half training samples coming from real, and from the buffer seems a bit arbitrary to me, especially combined with the choice of using only training samples for small t.
Wouldn't it make sense to use a ratio that varies more smoothly, i.e. an increasing proportion of buffer samples as t increases?

Also, the results in Table 1 look at the percentage of top-ranked predictions with RMSD smaller than 2 and 5A.
2A certainly is a commonly used and satisfactory threshold.
On the other hand, 5 is admittedly a very high threshold (cf the remark at the end of Section 4 stating that a RMSD larger than 4A constitutes a negative example).
Do you have any comment about this?

**Details Of Ethics Concerns:**

This is maybe a long shot and I honestly would be sad to see this submission being disqualified. This being said, as stated in my review:
One main concern I have is related to the relationship of the current manuscript with the already published DiffDock.
While the Confidence Bootstrapping is very specific to diffusion models and clearly builds on top of DiffDock, the authors have made a number of somewhat significant changes to it, even before testing the Confidence Bootstrapping (i.e. the results in Table 1).
I think it makes things a bit confusing for readers that are already very familiar with the field and the method (which I was not).
On a side note, while I do not think that the anonymity of this work has been technically breached, I think that this lack of clear separation between (Corso et al. 22) and the current manuscript gives hints that they are likely to have been authored by the same people.

---

> ### Author Response · Authors · 2023-11-18
> **Response to Reviewer kTFp - Part 1/2**
>
> Thank you very much for your constructive feedback and positive review. Below we respond to each of your questions and we have updated the manuscript to reflect these changes. We are glad you appreciated the paper and we hope that our responses clarify your questions.
>
> **Q: The authors have made a number of somewhat significant changes to it, even before testing the Confidence Bootstrapping (i.e. the results in Table 1). I think it makes things a bit confusing for readers that are already very familiar with the field and the method (which I was not).**
>
> Most of the changes we made were targeted at making running the score and confidence models faster to make our method faster. We tried to clarify these points in the paper and, as you suggested, to make contributions more clear we have added some rows in Table 1.
>
> **Q: Although the writing and presentation is overall very good, I find a few small parts to lack clarity.**
>
> Thank you for raising these points, we responded to each point below with an explanation, and we tried to clarify these points in the manuscript as well. Please let us know if you still find some of the explanations unclear.
>
> **1. For instance in Section 4.2 (paragraph 4), it is not really clear to me why the more local and easier task of testing the goodness of a pose can make it easier to generalize to unseen targets.**
>
> The harder an underlying task is, the more a model is likely to overfit by capturing some spurious correlations in the data instead of the underlying signal. Classifying poses is generally considered to be an easier task than generating new poses. This is why we hypothesized that the confidence model would generalize better to unseen targets compared to the score model.
>
> **2. The relationship to the multi-resolution structure of diffusion models is also not very clear to me.**
>
> At some noise level, the score model has to reason about what types of conformations are more likely to be correct, i.e. it should reason about what part of the space contains the optimal conformation(s). The larger the noise, the more the space over which the score model has to reason is large (since the noise level provides it a prior over the state space), and, therefore, the score model will rely on heuristics that are more likely to overfit.
>
> **3. The explanation and justification on the choice to limit the order of the spherical harmonics is a bit cryptic, especially compared to the level of details of the rest.**
>
> In all honesty, we believe this is a relatively unimportant empirical finding, that is purely aimed at reducing the runtime of the score and confidence models and not significantly affecting the way they work. We have clarified and put less emphasis on it in the updated manuscript to avoid potential confusion.
>
> **Q: To get a clearer overall picture of the performance of the proposed approach(es), I think it would make sense to add the Confidence Bootstrapping results in Table 1 (even if results would only be reported in the DockGen-clusters column).**
>
> Thank you for the suggestion, we added this row.
>
> **Q: In addition to be able to see more clearly the improvement in terms of performance, it would also be useful to see the runtime of this approach. I have no insight on how costly the confidence bootstrapping is, from a runtime perspective.**
>
> Thank you for raising this point. We added this discussion in Appendix C.3 and referred to it in the main text as well. The runtime of our approach depends on the number of complexes sampled and the number of gradient update steps. In our experiment, on average we sample 320 complexes in the cluster for every 200 gradient update steps (with a batch size of 5). We interleave these two operations 60 times. With these parameters and training on one Nvidia A6000 GPU, the average run time is 8 hours.
>
> One can try to compare this cost with for example running GNINA (here we assume the default version without extra search exhaustiveness), which generalizes better without retraining but it is considerably slower and cannot be finetuned on specific domains. Assuming the methods (GNINA & DiffDock w/ confidence bootstrapping) perform similarly, then one can ask in which setting would it be faster to run either one of them. It takes 232 inference complexes for DiffDock to amortize the cost of the finetuning and from then onwards it can provide significant runtime improvements. As many screening campaigns require significantly more than 232 complexes, we believe that the idea of finetuning a docking model to the specific target class of interest may prove a useful feature of these models in the future.

---

> ### Author Response · Authors · 2023-11-18
> **Response to Reviewer kTFp - Part 2/2**
>
> **Q: The choice, during the update step, of using a ratio of half-half training samples coming from real, and from the buffer seems a bit arbitrary to me, especially combined with the choice of using only training samples for small t. Wouldn't it make sense to use a ratio that varies more smoothly, i.e. an increasing proportion of buffer samples as t increases?**
>
> Thank you for raising these points. Regarding the use of training samples, while the RL objective is to directly maximize reward/confidence, we also want to prevent the model from overfitting into pathological spaces of the confidence model (where the confidence model gives high confidence for bad poses) aka "overoptimization", which is why we included the real samples from training set. We added a discussion on this point in Appendix D and we hope this clarifies the intuition of combining the two losses. We also added an ablation experiment that uses no real samples during fine-tuning (see Appendix E).
>
> Regarding the sampling ratio/weight of the training loss for different t. Note that because of the way we sample t (which is directly related to the loss weight $\lambda(t)$), even when considering $t_{min}=0.3$ the proportion of buffer samples is 0 until 0.3 and then increases gradually. We have clarified this point in the manuscript by instead of framing it in terms of sampling distribution we look at the difference in loss weight $\lambda$ vs $\lambda'$.
>
> **Q: Also, the results in Table 1 look at the percentage of top-ranked predictions with RMSD smaller than 2 and 5A. 2A certainly is a commonly used and satisfactory threshold. On the other hand, 5 is admittedly a very high threshold (cf the remark at the end of Section 4 stating that a RMSD larger than 4A constitutes a negative example). Do you have any comment about this?**
>
> Indeed 5A is large and a pose 5A away typically cannot be considered a "good pose", however, we report it (while definitely not focusing on this throughout the paper) because (1) it has been used a lot in the literature, (2) it can represent approximately whether or not the model guesses the correct pocket which might be a useful question when dealing with blind docking.
>
> ___
>
> Thank you again for your time and effort spent in reviewing and providing careful feedback, we hope to have clarified your questions!

---

> > ### Comment · Reviewer_kTFp · 2023-11-21
> >
> > Thanks to the authors for clearing some of my misunderstandings, answering in details to me concerns or questions and updating the manuscript accordingly.
> > I find this version of the manuscript to be improved and easier to read.
> > I am not going to increase my rating of the paper, which was already very high, though.

---

### Official Review · Reviewer_ksBz · 2023-11-06

**Soundness:** 3 good
**Presentation:** 3 good
**Contribution:** 3 good
**Rating:** 6
**Confidence:** 4

**Summary:**

A new docking benchmark called DockGen is introduced in this paper. Molecular docking methods based on machine learning do not perform well when confronted with unknown binding pockets, and this issue is not adequately addressed in previous benchmarks. In this benchmark, the validation and testing datasets are extracted from a different database, Binding MOAD, which contains new classes of binding pockets (based on clustering). The testing binding pockets are ensured to have a different shape than those in the training dataset. The ML-based docking methods fail in this benchmark, so a new training strategy is proposed to improve the results of the diffusion-based models. This strategy is called Confidence Bootstrapping and can be used to train (or fine-tune) DiffDock. The paper demonstrates that this methodology is effective in improving the results of DiffDock in the proposed benchmark, which focuses on model generalization.

**Strengths:**

- The motivation of this paper is very clear, and new benchmarks for ML-based docking are needed.
- The new benchmark is sourced from a different database than used in previous benchmarks, and the data is carefully curated.
- Poor performance of the known ML-based docking methods is proven using the new benchmark.
- A new method for training diffusion models is proposed to mitigate the problems with generalization.
- The proposed Confidence Bootstrapping method achieves good results for the introduced benchmark.
- Confidence Bootstrapping is implemented for DiffDock, where some architectural components (score and confidence models) were optimized.
- Two ways of expanding the training datasets are proposed, and they lead to better testing results.
- The benchmark and code were published along with this paper.

Overall, the idea behind the paper is clear, and the propositions are original and significant for the field of study (improving ML-based molecular docking).

**Weaknesses:**

- The mathematical formulas in the paper should be corrected. Some parentheses are missing, and there are some symbols that were not introduced in the text, e.g. $p_{0t}$. I understand that the equation in Section 4.1 should correspond to the equation introduced in the cited publication (Song et al., 2021), but the new undefined symbols and missing parentheses make this equation hardly readable.
- In Table 1, the results of DiffDock with modification explained in Section 4.4 and without dataset extension should also be presented.
- In this benchmark, the only evaluation metric (besides computation time) is the percentage of the poses with RMSD below 2 or 5 A. Given the recent criticism of ML-based docking models, it would be advised to include conformation quality metrics like those proposed in PoseBusters [1].
- It would be interesting to quantify the similarity between the binding pockets used in training, fine-tuning, and testing. I am curious if using complexes from the same dataset for fine-tuning and testing could create biases in the data due to the way the data is preprocessed and filtered. Furthermore, you should also try fine-tuning using only PDBBind structures, and see if the results for the DockGen benchmark improve in this setup.

[1] Buttenschoen, M., Morris, G. M., & Deane, C. M. (2023). PoseBusters: AI-based docking methods fail to generate physically valid poses or generalise to novel sequences. arXiv preprint arXiv:2308.05777.

Before this paper can be published, it is essential to clarify the mathematical formulation of the method. Furthermore, additional results that disentangle the three new factors (architectural changes, dataset expansion, and confidence bootstrapping) would not only increase the credibility of the findings but also enhance the overall quality of the paper. I am willing to increase my score if my comments are properly addressed by the Authors.

-----------
Edit: I changed my score (5 -> 6) after reading the other reviews and Authors' responses. The paper was significantly improved. Most of my concerns were resolved, and it seems the remaining concerns can be resolved in the final version of the manuscript if the running experiments are finished.

**Questions:**

1. Why have you decided to call your method Confidence Bootstrapping? This setup does not remind me of the classical bootstrapping method. The sampling is done from the continuous distribution (not a discrete distribution with replacement), and each sample is processed independently (not resampled as a subset of the population sample). Why is this method not called, e.g., Confidence Sampling?
2. What do you think about including conformation quality metrics in the benchmark, like those proposed in PoseBusters? Do you think that self-training using the confidence model can impact the pose quality in any significant way?
3. Based on the description in the paper, I was not sure which dataset was used for fine-tuning. I checked in the code, and `DistillationDataset` uses the Binding MOAD data. Could you confirm that? (see also the last point in Weaknesses)

---

> ### Author Response · Authors · 2023-11-18
> **Response to Reviewer ksBz - Part 1/2**
>
> Thank you very much for your thorough review, we truly appreciate your time and effort. We responded below to all your comments and questions and adapted the manuscript to reflect them.
>
> **Q: The mathematical formulas in the paper should be corrected. Some parentheses are missing, and there are some symbols that were not introduced in the text**
>
> Thank you for pointing this out. We fixed the equation and the symbols.
>
> **Q: In Table 1, the results of DiffDock with modification explained in Section 4.4 and without dataset extension should also be presented.**
>
> Thank you for the suggestion we added this row to the table. Unfortunately, the model has not finished training yet so we have put the performance after approximately 70% of the expected training time and we will update the result with the final performance.
>
> **Q: In this benchmark, the only evaluation metric (besides computation time) is the percentage of the poses with RMSD below 2 or 5 A. Given the recent criticism of ML-based docking models, it would be advised to include conformation quality metrics like those proposed in PoseBusters [1].**
>
> We agree with the reviewer and the authors of [1] that the quality of poses is an important metric. However, [1] shows that although the poses coming out of the diffusion model do not pass all quality metrics most of the time, they do pass them while remaining accurate typically if a quick energy minimization is applied. In [1] (Figure 5) the performance of DiffDock <2A goes down (in their benchmark) only from 38% to 35% if quality checks are also applied, assuming poses are relaxed (instead of 12% without relaxation). Therefore we believe that it makes sense to apply these checks in combination with the relaxation. To perform this test, we reached out (twice) to the authors of [1] to get the code used for the relaxation procedure, however, they said they will only release it in the future. Once this code is made available we'd be very interested in adding these metrics as the reviewer suggested!
>
> **Q: It would be interesting to quantify the similarity between the binding pockets used in training, fine-tuning, and testing. I am curious if using complexes from the same dataset for fine-tuning and testing could create biases in the data due to the way the data is preprocessed and filtered. Furthermore, you should also try fine-tuning using only PDBBind structures, and see if the results for the DockGen benchmark improve in this setup… Based on the description in the paper, I was not sure which dataset was used for fine-tuning. I checked in the code, and DistillationDataset uses the Binding MOAD data. Could you confirm that?**
>
> We are sorry that the manuscript was not clear on the datasets used for fine-tuning and testing, we have clarified it in the manuscript and we explain it in more detail below. Indeed for fine-tuning and testing, we use data from the BindingMOAD dataset as we are finetuning on protein clusters that were not present in PDBBind. The critical difference is that the fine-tuning process only uses the protein structures and the ligand identities and not the docked structures that are then what is evaluated at testing time. Further, there was also no leakage w.r.t. the hyperparameters as these were obtained when running fine-tuning on the validation set (that contains different clusters). This setup simulates a relatively common setting in screening campaigns where researchers typically have access to some accurate protein structures and the identity of some ligands binding to it derived from some experimental assays. At that point, before running a large-scale screening campaign the researchers could use confidence bootstrapping to finetune their docking model.
>
> Thank you for the suggestion on quantifying the similarity between binding pockets in the training data vs testing. We conducted a careful analysis of the similarity of the binding sites in DockGen test set and the training set and comparing this with the similarities observed in the commonly used PDBBind splits. We extract the binding sites of different complexes and then compare them with one another both at a sequence/chemical (using the BLOSUM62 amino acid substitution matrix to quantify similarity of aligned residues) and structural (using the TMalign output for any aligned residues) level. The results, shown in Figure 5, show significant overlap in the pockets in PDBBind and very little similarity when looking at DockGen pockets. More details on this analysis can be found in Appendix A.2.

---

> ### Author Response · Authors · 2023-11-18
> **Response to Reviewer ksBz - Part 2/2**
>
> **Q: Why have you decided to call your method Confidence Bootstrapping? This setup does not remind me of the classical bootstrapping method.**
>
> We agree with the reviewer that the method is not related to the method of bootstrapping used in mathematics. However, the term bootstrapping has a wider meaning in English where it indicates "to create something using the minimum amount of resources possible" (Oxford dictionary). In this sense our motivation for choosing the name is the ability of the model to learn to sample good poses for unseen protein classes without access to data only relying on the feedback from the confidence model and the binary information that a protein and ligand bind (the minimum amount of resources possible).
>
> **Q: What do you think about including conformation quality metrics in the benchmark, like those proposed in PoseBusters? Do you think that self-training using the confidence model can impact the pose quality in any significant way?**
>
> See the answer above. Moreover, including relaxation and quality checks could improve the confidence score and therefore lead to the sampling of complexes that are more likely to be correct and can be relaxed to good poses (in terms of quality metrics).
>
> ___
>
> Thank you again for the thoughtful feedback and questions! We hope the improvements and clarifications might warrant raising your score and we are happy to continue the conversation.

---

### Meta-Review · Area_Chair_1wG5 · 2023-12-06

**Metareview:**

The paper makes two contributions to molecular docking using machine learning. First, the paper introduces DockGen, a novel benchmark that focuses on the generalizability of different methods to novel protein binding poses. The benchmark exposes a limitation of machine learning models that underperform search-based methods such as SMINA and GMINA. Beyond that, the paper introduces an interesting self-training method for the problem that not only delivers promising results but is also well-motivated from the principle that it is easier to evaluate a pose than generate one. After an internal discussion, all reviewers were supportive of accepting the paper. In light of these contributions and the positive outcome of the rebuttal phase, it is my pleasure to recommend acceptance of the paper.

**Justification For Why Not Higher Score:**

The reviewers had some less critical reservations about lack of additional results.

**Justification For Why Not Lower Score:**

The paper makes two significant contributions that are technically sound and novel.

---

### Decision · Program_Chairs · 2024-01-16

Accept (poster)